# Transcriptome Analysis of *mfs2*-Defective *Penicillium digitatum* Mutant to Reveal Importance of Pd*mfs2* in Developing Fungal Prochloraz Resistance

**DOI:** 10.3390/microorganisms12050888

**Published:** 2024-04-28

**Authors:** Rongrong Cuan, Shaoting Liu, Chuanyou Zhou, Shengqiang Wang, Yongliang Zheng, Yongze Yuan

**Affiliations:** 1Hubei Key Laboratory of Genetic Regulation and Integrative Biology, School of Life Sciences, Central China Normal University, Wuhan 430079, China; crr@mails.ccnu.edu.cn (R.C.); zhouchuanyou@mails.ccnu.edu.cn (C.Z.); shqwang@ccnu.edu.cn (S.W.); 2School of Political and Law, Huanggang Normal University, Huanggang 438000, China; liusting-777@hgnu.edu.cn; 3Hubei Key Laboratory of Economic Forest Germplasm Improvement and Resources Comprehensive Utilization & Hubei Collaborative Innovation Center for the Characteristic Resources Exploitation of Dabie Mountains, Huanggang Normal University, Huanggang 438000, China; ylzheng@hgnu.edu.cn

**Keywords:** transcriptome, *Penicillium digitatum*, major facilitator superfamily (MFS), *mfs2*-defective strain, prochloraz resistance

## Abstract

Demethylation inhibitors (DMIs), including prochloraz, are popular fungicides to control citrus postharvest pathogens such as *Penicillium digitatum* (green mold). However, many *P. digitatum* strains have developed prochloraz resistance, which decreases drug efficacy. Specific major facilitator superfamily (MFS) transporter gene *mfs2*, encoding drug-efflux pump protein MFS2, has been identified in *P. digitatum* strain F6 (PdF6) to confer fungal strain prochloraz resistance. However, except for the drug-efflux pump function of MFS2, other mechanisms relating to the Pd*mfs2* are not fully clear. The present study reported a transcriptome investigation on the *mfs2*-defective *P. digitatum* strain. Comparing to the wild-type strain, the *mfs2*-defective strain showed 717 differentially expressed genes (DEGs) without prochloraz induction, and 1221 DEGs with prochloraz induction. The obtained DEGs included multiple isoforms of MFS transporter-encoding genes, ATP-binding cassette (ABC) transporter-encoding genes, and multidrug and toxic compound extrusion (MATE) family protein-encoding genes. Many of these putative drug-efflux pump protein-encoding genes had significantly lower transcript abundances in the *mfs2*-defective *P. digitatum* strain at prochloraz induction, as compared to the wild-type strain, including twenty-two MFS transporter-encoding genes (*MFS1* to *MFS22*), two ABC transporter-encoding genes (*ABC1* and *ABC2*), and three MATE protein-encoding genes (*MATE1* to *MATE3*). The prochloraz induction on special drug-efflux pump protein genes in the wild-type strain was not observed in the *mfs2*-defective strain, including *MFS21*, *MFS22*, *ABC2*, *MATE1*, *MATE2*, and *MATE3*. On the other hand, the up-regulation of other drug-efflux pump protein genes in the *mfs2*-defective strain cannot recover the fungal prochloraz resistance, including *MFS23*, *MFS26*, *MFS27*, *MFS31*, *MFS33*, and *ABC3* to *ABC8*. The functional enrichment of DEGs based on Kyoto Encyclopedia of Genes and Genomes (KEGG), Clusters of Orthologous Groups (COG), and euKaryotic Orthologous Groups (KOG) database resources suggested some essential contributors to the *mfs2*-relating prochloraz resistance, including ribosome biosynthesis-related genes, oxidative phosphorylation genes, steroid biosynthesis-related genes, fatty acid and lipid metabolism-related genes, and carbon- and nitrogen-metabolism-related genes. The results indicated that the MFS2 transporter might be involved in the regulation of multiple drug-efflux pump protein gene expressions and multiple metabolism-related gene expressions, thus playing an important role in developing *P. digitatum* prochloraz resistance.

## 1. Introduction

Postharvest citrus fruits are usually infected by *Penicillium digitatum* pathogens during storing and transporting processes, and, as such, green mold disease significantly reduces citrus fruit production in markets [1,2]. A large number of chemical drugs (fungicides) have been applied to control green mold, including demethylation inhibitors (DMI) class fungicides. Currently, among the common classes of antifungal drugs with specific targets, the DMI class of fungicides is considered more suitable to inhibit phytopathogenic *Penicillium* ssp., especially *P. digitatum* [3,4,5]. DMI fungicides including triadimefon, imazalil and prochloraz, all of which target the key step in the biosynthesis of fungal ergosterol (i.e., the lanosterol 14α-demethylation). However, the long time required by such DMI-fungicide treatments led to increasing the efforts to develop drug-resistant fungal strains in the field. Regarding green mold control, the high frequency needed to develop triadimefon- and imazalil-resistant *P. digitatum* isolates in storing and transporting conditions has undesirably lowered the control efficacy of these two DMI-fungicides [6,7]. Now prochloraz, a chemical compound of azole DMI, is still widely used in the green mold control in China’s citrus industry chains, as result of its economic cost-effectiveness [5]. Nevertheless, *P. digitatum* strains with high resistance to prochloraz have emerged, which brought about more attention to the underlying mechanisms. One high prochloraz-resistant strain of *P. digitatum* has been isolated and characterized in our laboratory, and through gene-knockout and complementation experiments, a drug efflux-pump protein-encoding gene, Pd*mfs2*, has been identified to be an important contributor to the fungal prochloraz resistance [8]. Based on these fungal materials, more mechanisms underlying the Pd*mfs2*-relating prochloraz resistance can be further studied.

Fungal resistance to DMI fungicides including prochloraz is usually developed from several major mechanisms. The first is the over-expression of fungicide-targeting proteins or enzymes such as sterol 14α-demethylase, which is encoded by the gene *erg11* (i.e., *cyp51*) [9]. The other mechanism regarding the gene *cyp51* has been known as gene mutations in its coding sequence or promoter region, including a 199-bp insertion [7,10,11], specific ‘CC’ insertion [12], and point mutations [13]. Besides the gene *cyp51*-targeted mechanisms, the over-expression of genes encoding drug efflux-pump proteins can be an essential strategy for pathogenic fungi to develop resistance against various fungicides including prochloraz. Such drug efflux-pump proteins can be classified into three superfamilies: (1) the major facilitator superfamily (MFS), (2) ATP-binding cassette (ABC) superfamily, and (3) multidrug and toxic compound extrusion (MATE) transporter superfamily. In fungal cells, these drug efflux-pump proteins are responsible for exporting fungicide(s) out of membrane to decrease intracellular drug concentration to induce fungicide resistance [14,15]. Some of the drug efflux-pump proteins, especially MFS and ABC superfamily members, also serve as comprehensive metabolism-relating transporters with multiple physiological functions. MFS transporters have been verified as secondary active transporters to produce ion gradients that are directly associated with cellular energy metabolisms such as oxidative phosphorylation [16,17]. Such MFS subfamily members also function as drug H+ antiporters to develop fungal multidrug resistance [18,19,20]. MFS transporters are also extensively involved in fungal virulence to their hosts, especially at no fungicide conditions [21,22,23]. On the other hand, at fungicide treatments, over-expression of specific MFS transporters led to antifungal resistance [24]. On the contrary, the knockout of MFS-encoding gene(s) decreased fungal resistance to fungicide(s) including prochloraz [8,23,25]. Besides MFS, ABC transporter superfamily genes also contribute to fungal resistance at fungicide treatments. ABC genes including *ABC1*, *ABC2*, and *ABC3* are extensively involved in fungal resistance to prochloraz, both in *P. digitatum* and in *P. italicum* [9,26]. The simultaneous over-expression of MFS and ABC genes has been found in the highly prochloraz-resistant fungus including *Candida* spp. isolates, *P. digitatum* strain HS-F6 and *P. italicum* strain YN1 [9,26,27,28]. Unlike MFS and ABC, MATE superfamily transporters are usually associated with bacterial antibiotic resistance [29], and in contrast, contribute to fungal drug resistance in a few cases [30,31].

Genomics and RNA-seq studies revealed multiple iso-genes encoding MFS and ABC transporters as contributors to developing fungicide resistance. *P. digitatum* genomes (i.e., Pd1 and PdW03 genomes) have been identified to contain more than 80 locus encoding MFS-type transporter proteins [32,33]. Some of these MFS-encoding genes were responsible for the fungal resistance to chemical fungicides, including Pd*Mfs1* in the azole- or DMI-resistant *P. digitatum* strain PdW03 [25], Pd*mfs2* in the prochloraz-resistant *P. digitatum* strain PdHS-F6 [8], and Pd*MFS1* in the multidrug-resistant *P. digitatum* strain Pd1 [20,34]. Genomic studies also showed that multiple copies and chromosomal locations of ABC transporter proteins are highly associated with fungicide resistance in the *P. digitatum* strains, including Pd01-ZJU [35], Pd1 [32], and PdW03 [33]. RNA-seq evidence has suggested simultaneous up- or down-regulation of specific MFS and ABC transporter genes in *P. digitatum* strains to develop fungicide resistance [9,26,36]. Such transcriptomic responses were also reported in many other pathogenic fungi with fungicide resistance phenotypes [28,37,38,39]. Considering those genes encoding drug-pump proteins, usually as MFS-, ABC-, and MATE-type transporters, all of which exhibited multiple isoforms in the *P. digitatum* genomes, the underlying mechanisms to develop fungicide resistance need further investigation.

On the other hand, fungi can make adaptive responses to fungicide stress conditions by changing metabolism-relating gene expression patterns. These metabolisms and cellular stimuli processes in response to specific fungicide(s) have been studied, including ergosterol biosynthesis pathways, lipid and fatty acid oxidation pathways, cell wall maintenance, oxidative-stress-responsive processes, carbohydrate and amino acid metabolisms, cellular energy metabolisms, post-translational modification processes, and signal transduction pathways. All these pathways are highly dependent on many stress-responsive genes, including various ERG-encoding genes (such as *erg1*, *erg3*, *erg11*, *erg24*, *erg25* and so on) [36,40,41,42], acetyl-CoA carboxylase (ACCase)-encoding genes [43,44,45,46,47], reactive oxygen species (ROS)-metabolizing enzyme-encoding genes [48,49,50,51], mitochondrial respiratory chain protein-encoding genes [52,53,54,55], ubiquitin-encoding genes [56,57,58,59], and a series of protein kinase-encoding genes involved in mitogen-activated signal transductions [55,60,61]. As reported, MFS and ABC transporters played multiple roles in the transport of a diverse range of metabolic substrates and intermediates [62,63]. *P. digitatum* MFS transporters can display different roles during pathogen–fruit interaction [20]. It would be necessary to explore the association of MFS transporter(s) with the sophisticated metabolic responses in developing fungicide resistance.

The present study investigated transcriptomic changes between two *P. digitatum* strains, i.e., PdF6 and PdF6Δ*mfs2* at prochloraz induction or no prochloraz induction, using RNA-sequencing and the following differentially expressed gene identification and enrichments based on COG, KOG, and KEGG databases to show the importance of gene Pd*mfs2* in *P. digitatum* prochloraz resistance as well as the relating metabolism backgrounds.

## 2. Materials and Methods

### 2.1. Strains and Media

The *P. digitatum* strain used in this study was previously isolated by our research group [64], which was highly resistant to DMI-fungicide prochloraz with an EC_50_ value of 7.90 mg·L^−1^. Meanwhile, the *mfs2*-deleted *P. digitatum* strain (PdF6Δ*mfs2*) was less resistant to prochloraz with an EC_50_ value of 6.80 mg·L^−1^ [8]. In the present study, the prochloraz resistance of these two *P. digitatum* strains was verified using methods described before [8,64]. *P. digitatum* strains were cultivated on potato dextrose agar (PDA) medium (extract of 200 g potato boiled water, 20 g dextrose, and 15 g agar per liter) at 28 °C for 5 days to prepare respective conidial suspension (10^7^ spores mL^−1^) as previously described [26]. Then, 20 μL of conidial suspension (10^7^ spores mL^−1^) of each strain was cultured in potato dextrose broth (PDB) medium at 28 °C for about 2 days. Prochloraz induction experiment was carried out for sample preparation. Prochloraz at the concentration of EC_50_ (7.90 mg·L^−1^ for wild-type strain and 6.80 mg·L^−1^ for *mfs2*-deleted strain) was added to 100 mL PDB medium with 180 rpm shaking for an extra 6 h of growth at 28 °C. The mycelia were filtered and washed several times using double distilled water. In the present study, 4 samples in total were used for the following RNA manipulations: prochloraz-induced and not induced wild-type *P. digitatum* strain (designated as Pd-wt-I and Pd-wt-NI, respectively); and prochloraz-induced and not induced *mfs2*-deleted *P. digitatum* strain (designated as Pd-d-I and Pd-d-NI, respectively).

### 2.2. RNA Extraction, RNA-seq Library Construction and Illumina Sequencing

Total RNA was extracted using RNAiso Plus (TaKaRa Biotech. Co., Dalian, China) according to the manufacturer’s protocol. All RNA samples were treated with DNase I (TaKaRa Biotech. Co., Dalian, China). RNA degradation and contamination were monitored on 1% agarose gels. RNA purity was checked using the Nano-Photometer^®^ spectrophotometer (IMPLEN, Westlake Village, CA, USA). RNA concentration was measured using Qubit^®^ RNA Assay Kit in Qubit^®^ 2.0 Flurometer (Life Technologies, San Francisco, CA, USA). RNA integrity was assessed using the RNA Nano 6000 Assay Kit of the Bioanalyzer 2100 system (Agilent Technologies, Santa Clara, CA, USA). From each sample, 3 μg of total RNA was taken to construct strand-specific cDNA libraries using the NEBNext^®^ Ultra™ RNA Library Prep Kit for Illumina^®^ (NEB, Ipswich, MA, USA) following the manufacturer’s instructions. The required fragments were enriched by PCR amplification, and the products were purified using AMPure XP system (Beckman Coulter, Brea, CA, USA). Library quality was assessed on the Agilent Bioanalyzer 2100 system. After cluster generation, the library preparations were sequenced on an Illumina Hiseq 2000 platform (Illumina, San Diego, CA, USA) and the resulting paired-end reads (raw reads) in ~150 bp length were deposited for further analysis.

### 2.3. Assembly of Reads and Unigenes and Analysis of SNP Sites

Prior to sequence analysis, high-quality clean reads were obtained by removing reads containing adapter and low-quality reads. Transcriptome data and reference genome sequence alignment was accomplished by the HISAT platform, a highly efficient system for aligning reads from RNA sequencing [65]. All pair-end clean reads were aligned to reference genome *Penicillium digitatum* Pd1 (GenBank accession number: GCA_000315645.2). Then, the aligned reads of each sample were assembled by StringTie methods [66] to obtain transcripts and unigenes. The alignment efficiency was estimated by the percentage of mapped reads, uni-mapped reads, and multiple-mapped reads to the total clean reads. Prior to differentially expressed gene analysis, the read counts were adjusted according to one scaling normalized factor for each library using edgeR program packages [67]. Based on the alignment results of each sample against the reference genome, single-base mismatches between the sequenced samples and reference genome were identified to recognize potential single-nucleotide polymorphisms (SNP) sites using the GATK method [68]. The two types of SNP sites (i.e., transition sites and transversion sites), according to different base substitution styles, were both assessed in the present study.

### 2.4. Analysis of Differentially Expressed Genes (DEGs)

In order to reflect transcript abundance in the present RNA-sequencing, the number of mapped reads in the samples and the length of transcripts both required normalization, i.e., gene expression quantification. The Fragments Per Kilobase of transcript per Million fragments mapped (FPKM) was used to measure transcript or gene expression levels according to the statistics methods described before [69]. Differentially expressed genes (DEGs) analysis was performed using the DEGSeq R package (1.20.0) to calculate fold-changes in the expression level for each gene, defined as the ratio of the FPKM values. The *p*-values were statistically corrected by using the Benjamini–Hochberg method to assess the significance for the differences in transcript abundance [70]. The cut-off value to define differentially expressed genes (DEGs) was the adjusted *p*-value ≤ 0.05 and at least 2-fold change (i.e., the absolute value of log2 Fold Change (log2FC) ≥ 1.0) in transcript abundance between two comparison samples. The identified DEGs were hierarchically clustered by Cluster 3.0 [71], and then subjected to heat-map analysis by Plotly 4.0 (Montreal, Quebec) software. The distribution of up- and down-regulated DEGs versus unchanged unigenes was visualized using Volcano plots [72]. All the DEGs were annotated, and then functionally classified and enriched according to the three common databases, including the Clusters of Orthologous Groups (COG) database (http://www.ncbi.nlm.nih.gov/COG/; 25 November 2020) [73], clusters of euKaryotic Orthologous Groups (KOG) database (http://www.ncbi.nlm.nih.gov/KOG/; 25 November 2020) [73], and Kyoto Encyclopedia of Genes and Genomes (KEGG) database (http://www.genome.jp/kegg/; 1 January 2022) [74]. The KOBAS software (version 3.0) was applied to perform the COG-, KOG-, and KEGG-based classification and enrichment of the present DEGs according to the method of Mao et al. [75].

### 2.5. Validation of DEGs with Quantitative Real-Time PCR (qRT-PCR)

Quantitative real-time PCR (qRT-PCR) was carried out to validate the expression patterns of DEGs, including genes encoding MFS, ABC and MATE drug-pump proteins and the other fungicide resistance genes involved in multiple cellular metabolic processes. RNA samples were collected independently from RNA-seq experiments, and the first-strand cDNA was generated using PrimeScript^TM^RT reagent Kit with gDNA Eraser (TaKaRa, Dalian, China) according to the manufacturer’s instructions. qRT-PCR reactions were conducted in a BIO-RAD CFX96 qPCR system using SYBR Premix Ex Taq™ II kits (Takara, Dalian, China). The primers in the present study were designed using the software Primer Premier 5.0, as listed in Appendix A, and their quality in the qRT-PCR amplification was evaluated by melting curve analysis. All qRT-PCR reactions were performed with three technical replicates and the thermal conditions were as follows: 30 s at 95 °C, followed by 35 cycles of 20 s at 95 °C, 30 s at 58 °C and 30 s at 72 °C. The relative quantification of each gene expression level was normalized according to the β-actin gene expression and calculated from the threshold cycle according to the 2^−ΔΔCt^ method.

## 3. Results

### 3.1. Transcriptome Sequencing and Reads Assembly

In the present study, wild-type and *mfs2*-deleted *P. digitatum* strains were treated with or without DMI-fungicide prochloraz to prepare four RNA-seq samples, including Pd-d-I (i.e., mfs2-deleted *P. digitatum* strain with prochloraz induction), Pd-d-NI (i.e., *mfs2*-deleted *P. digitatum* strain with no prochloraz induction), Pd-wt-I (i.e., wild-type *P. digitatum* strain with prochloraz induction), and Pd-wt-NI (i.e., wild-type *P. digitatum* strain with no prochloraz induction). The Illumina sequencing data are summarized in Appendix A. The four transcriptomic libraries contained 6,536,999,320, 7,141,474,368, 6,966,924,166, and 6,621,791,146 clean bases, respectively. By removing adaptor sequences and undesirable reads including ambiguous, low-quality, and duplicated sequence reads, 21,847,213, 23,870,541, 23,294,756 and 22,149,643 clean reads were generated from the four libraries with Q30 > 90%, and the GC content of the four libraries was around 50%. These results suggested high quality for the present RNA-sequencing.

The percentages of reads mapping to the reference genome in different samples are shown in Appendix A. Total reads in the sample Pd-d-I, Pd-d-NI, Pd-wt-I and Pd-wt-NI were 43,694,426, 47,741,082, 46,589,512 and 44,299,286, and 95.90%, 94.03%, 87.74% and 94.04% of the total reads were mapped to the reference genome Pd1 for the above four samples, respectively. In addition, the uniquely mapped reads occupied 93.79%, 92.38%, 86.36% and 91.98% of the total mapped reads for the four samples, respectively. In contrast, multiple mapped reads accounted for only a small fraction, that is, 2.12%, 1.64%, 1.38%, and 2.06% for the four samples, respectively. Appendix A also showed high equivalence between the reads mapped to plus and minus strand of genome sequence. These results indicated high quality of mapping analysis based on the Pd1 reference genome.

The SNP analysis for the four samples was summarized in Appendix A. The SNP numbers were 2458, 2527, 2311, and 2580 for the four samples, respectively. In detail, the number of Genic SNP in the genetic region was 1592, 1609, 1511, and 1675, respectively, and the number of Intergenic SNP in the intergenic region was 866, 918, 800, and 905, respectively. In addition, the percentage of the number of transitional SNP in the total number of SNP was 61.92%, 61.65%, 62.48%, and 61.94%, respectively. In addition, the percentages of transversion and heterozygosity in the total number of SNP were similar, i.e., both between 30% and 40%. These results indicated little change in *P. digitatum* genome structures with the gene *mfs2* deletion.

### 3.2. Analysis of Differentially Expressed Genes (DEGs)

According to the criteria *p*-value ≤ 0.05 and the absolute value of fold change ≥ 1.0, the present study totally identified 460 DEGs in wild-type *P. digitatum* strain with prochloraz induction compared to that with no prochloraz induction (i.e., Pd-wt-(I/NI)) (Appendix A). As shown in Appendix A, there were 147 DEGs in the *mfs2*-deleted *P. digitatum* strain with prochloraz induction compared to that with no prochloraz induction (i.e., Pd-d-(I/NI)); 1221 DEGs in the *mfs2*-deleted *P. digitatum* strain with prochloraz induction compared to wild-type *P. digitatum* strain with prochloraz induction (i.e., I-(Pd-d/Pd-wt)); and 717 DEGs in the *mfs2*-deleted *P. digitatum* strain with no prochloraz induction compared to wild-type *P. digitatum* strain with no prochloraz induction (i.e., NI-(Pd-d/Pd-wt)).

DEGs were annotated by alignment analysis using the common public databases. In total, 427, 128, 1169 and 676 DEGs in Pd-wt-(I/NI), Pd-d-(I/NI), I-(Pd-d/Pd-wt) and NI-(Pd-d/Pd-wt) were functionally annotated in the eight databases (Appendix A). For the above four comparative groups, there were 147, 32, 409 and 258 DEGs annotated in the COG database (http://www.ncbi.nlm.nih.gov/COG/; 25 November 2020), respectively; there were 170, 35, 486 and 288 DEGs annotated in the KOG database (https://www.ncbi.nlm.nih.gov/research/cog-project/; 25 November 2020), respectively; there were 243, 68, 689 and 408 DEGs annotated in the GO database (http://www.geneontology.org/; 1 July 2019), respectively; there were 73, 18, 299 and 162 DEGs annotated in the KEGG database (http://www.genome.jp/kegg/; 1 January 2022), respectively; there were 258, 65, 750 and 434 DEGs annotated in the Pfam database (http://pfam.xfam.org/; 1 January 2021), respectively; there were 196, 51, 586 and 346 DEGs annotated in the Swiss-Prot database (http://www.uniprot.org/; 1 January 2020), respectively; there were 364, 104, 1044 and 577 DEGs annotated in the eggNOG database (http://eggnogdb.embl.de/download/emapperdb-5.0.0/; 19 March 2019), respectively; and there were 426, 128, 1167 and 676 DEGs annotated in the NR database (https://ftp.ncbi.nlm.nih.gov/blast/db/; 30 July 2021), respectively.

### 3.3. DEG Analysis between the Wild-Type and mfs2-Deleted P. digitatum Strains at No Prochloraz Induction

The volcano plot analysis identified 717 DEGs in the *mfs2*-deleted strain as compared to the control, including 366 up-regulated and 351 down-regulated (Figure 1A). All unigene expression levels were determined by the FPKM values, and based on these values, a hierarchical cluster (i.e., heat map) analysis was performed to visualize DEG profiles between the wild-type and *mfs2*-deleted *P. digitatum* strains (Figure 1B). Further, KEGG enrichments classified the 366 down-regulated DEGs into 42 pathways with two significant enrichments, i.e., ‘pentose and glucuronate interconversions’ (ko00040) and ‘starch and sucrose metabolism’ (ko00500) (Figure 1C). In contrast, the 351 up-regulated DEGs were classified by KEGG enrichment into one significantly enriched pathway in the total 44 pathways, i.e., ‘nitrogen metabolism’ (ko00910) (Figure 1D).

According to the *q*-values, the top 3 significantly KEGG pathways, as well as the DEGs involved, were listed in Table 1, including ‘pentose and glucuronate interconversions’ (ko00040), ‘starch and sucrose metabolism’ (ko00500), and ‘peroxisome’ (ko04146). The DEGs in the three KEGG pathways included dihydrodipicolinate synthetase-encoding gene, exopolygalacturonase-encoding gene, exo-β-1,3-glucanase-encoding gene, α-L-rhamnosidase-encoding gene, fatty acyl-CoA oxidase-encoding gene, and carnitine acetyl transferase-encoding gene. In addition, energy-metabolism-related genes including ATP synthase-coding gene, NADH dehydrogenase-encoding gene, and cytochrome b-encoding gene were enriched into the KEGG pathway ‘Oxidative phosphorylation’ (ko00190). All the DEGs in the four KEGG pathways were down-regulated in the comparison NI-(Pd-d/Pd-wt). On the other hand, some DEGs were up-regulated in the comparison NI-(Pd-d/Pd-wt), which is also summarized in Table 1. These up-regulated genes responsive to the Pd*mfs2* knockout at no prochloraz treatment were mainly enriched in another four KEGG pathways associated with nitrogen and amino acid metabolisms, including nitrogen metabolism (ko00910), ‘tyrosine metabolism’ (ko00350), ‘phenylalanine metabolism’ (ko00360), and ‘tryptophan metabolism’ (ko00380).

Function classification of down-regulated DEGs between the wild-type and *mfs2*-deleted *P. digitatum* strains at no prochloraz induction was performed based on the COG and KOG databases. Both COG and KOG enrichments showed that these DEGs were mainly involved in ‘general function prediction only’ (R), ‘amino acid transport and metabolism’ (E) and ‘carbohydrate transport and metabolism’ (G) (Figure 2). As listed in Table 2, some of the DEGs were enriched in more than one COG or KOG class, including dihydrodipicolinate synthetase-encoding gene enriched in both ‘amino acid transport and metabolism’ (E) and ‘cell wall/membrane/envelope biogenesis’ (M) COG classes, α-L-Rhamnosidase-encoding gene enriched in ‘carbohydrate transport and metabolism’ (G) class in both COG and KOG databases, peroxin-encoding gene enriched in ‘general function prediction only’ (R) COG class and ‘intracellular trafficking, secretion, and vesicular transport’ (U) KOG class.

### 3.4. DEG Analysis in the Wild-Type and mfs2-Deleted P. digitatum Strains with Prochloraz Induction

The comparison of the wild-type and *mfs2*-deleted *P. digitatum* strains with prochloraz induction identified 460 and 147 DEGs, respectively, including 240 and 50 up-regulated genes and 220 and 97 down-regulated genes (Figure 3). Heatmap analysis was performed between strains with prochloraz induction and with no prochloraz induction (Figure 4).

In the comparison Pd-wt-(I/NI), up-regulated DEGs were significantly enriched in ‘oxidative phosphorylation’ (ko00190) and ‘steroid biosynthesis’ (ko00100) pathways (Figure 5A). These DEGs are listed in Table 3, including cytochrome c oxidase-encoding gene, NADH dehydrogenase-encoding gene, cytochrome b-encoding gene, ATP synthase-encoding gene and Erg-encoding genes. In addition, DEGs enriched in other KEGG pathways are also listed in Table 3. In the comparison Pd-d-(I/NI), up-regulated DEGs were significantly enriched in seven KEGG pathways, including three lipid metabolic pathways (i.e., ‘ether lipid metabolism’ (ko00565), ‘inositol phosphate metabolism’ (ko00562) and ‘glycerophospholipid metabolism’ (ko00564)), two carbohydrate metabolic pathways (i.e., ‘galactose metabolism’ (ko00052) and ‘starch and sucrose metabolism’ (ko00500)), ‘biosynthesis of antibiotics’ (ko01130), and the most significant pathway, ‘steroid biosynthesis’ (ko00100) (Figure 5B). The DEGs involved in these KEGG pathways are listed in Table 4. Among these DEGs, Erg25-encoding gene was up-regulated in Pd-wt-(I/NI) and Pd-d-(I/NI). In contrast, cytochrome c oxidase-encoding gene and NADH dehydrogenase-encoding gene were down-regulated in both comparisons.

In the comparison Pd-wt-(I/NI), the most up-regulated DEGs were enriched in the following KOG classes: 1) ‘general function prediction only’ (R), including 20 DEGs such as NADH-quinone oxidoreductase-encoding gene and aldehyde reductase-encoding gene; and 2) ‘energy production and conversion’ (C), including 18 DEGs such as NADH dehydrogenase-encoding gene, glyoxylate reductase-encoding gene, and cytochrome c oxidase-encoding gene. In the comparison Pd-d-(I/NI), the most up-regulated DEGs were enriched in the following KOG classes: (1) ‘general function prediction only’ (R) with only 4 hypothetical protein-encoding genes; and (2) ‘lipid transport and metabolism’ (I) and ‘secondary metabolites biosynthesis, transport and catabolism’ (Q) (Figure 6 and Table 5).

### 3.5. DEG Analysis between the Wild-Type and mfs2-Deleted P. digitatum Strains at Prochloraz Induction

The volcano plot showed the distribution of DEGs between the wild-type and *mfs2*-deleted *P. digitatum* strains at prochloraz induction (Figure 7A), including 608 up-regulated DEGs and 613 down-regulated DEGs. All the unigene expression levels were also determined by FPKM values, and based on these values, the heat map analysis was performed to visualize DEG profiles between the wild-type and *mfs2*-deleted *P. digitatum* strains at prochloraz induction (Figure 7B). Further, KEGG enrichment was performed to classify the DEGs in the mfs2-deleted P. digitatum strains at prochloraz induction. Among the down-regulated DEGs (Figure 7C), there were 20 DEGs significantly enriched in the KEGG pathway ‘ribosome’ (ko03010), including ribosomal protein-encoding gene and acidic ribosomal phosphoprotein-encoding gene. There were 17 DEGs enriched in another KEGG pathway ‘oxidative phosphorylation’ (ko00190), including cytochrome c oxidase-encoding gene, ATP synthase-encoding gene, NADH dehydrogenase-encoding gene, and cytochrome b-encoding gene. Generally, as shown in Figure 7D, the up-regulated DEGs were enriched in the KEGG pathways involved in fungal growth, lipid and fatty acid biosynthesis, and nitrogen-containing nutrient metabolisms, including ‘fatty acid biosynthesis’ (ko00061) and ‘nitrogen metabolism’ (ko00910). The DEGs enriched in the above KEGG pathways are listed in Table 6.

KOG-based annotation and functional classification confirmed the results of KEGG enrichments in the comparison I-(Pd-d/Pd-wt). The deletion of *mfs2* led to the down-regulation of many genes in the *P. digitatum* strain. These down-regulated DEGs were mainly classified into seven KOG classes, including ‘general function prediction only’ (R), ‘energy production and conversion’ (C), ‘translation, ribosomal structure and biogenesis’ (J), ‘amino acid transport and metabolism’ (E), ‘carbohydrate transport and metabolism’ (G), ‘post-translational modification, protein turnover, chaperons’ (O), and ‘secondary metabolites biosynthesis, transport and catabolism’ (Q) (Figure 8A). Meanwhile, the up-regulated DEGs in the comparison I-(Pd-d/Pd-wt) were classified into the similar KOG classes, as compared to the down-regulated DEGs (Figure 8B). However, the gene number of the up-regulated DEGs was lower than that of down-regulated DEGs in the four KOG classes (i.e., KOG class C, E, G, and J). In contrast, the gene number of the up-regulated DEGs was higher than that of down-regulated DEGs in the three KOG classes (i.e., KOG class K, O, and T). Such a difference in DEG distribution in the KOG classes might reflect different mechanisms to develop prochloraz resistance. The DEGs involved in the above KOG functional classifications are listed in Table 7. Actually, the down-regulated DEGs in the comparison I-(Pd-d/Pd-wt) did function in the ‘translation, ribosomal structure and biogenesis’ and ‘energy production and conversion’, including ribosomal protein-encoding genes, cytochrome c oxidase-encoding gene, ATP synthase-encoding gene, NADH dehydrogenase-encoding gene, and cytochrome b-encoding gene. The simultaneous down-regulation of these genes with gene Pd*mfs2* knockout indicated their potential correlation with Pd*mfs2*, i.e., their important roles in developing prochloraz resistance. On the other hand, the up-regulated DEGs in the comparison I-(Pd-d/Pd-wt) did function in the ‘transcription’, ‘post-translational modification, protein turnover, chaperons’, and ‘signal transduction mechanisms’, including specific transcription factor-encoding genes, ubiquitin carboxyl-terminal hydrolase-encoding gene, and MAPKKK-encoding gene. The up-regulation of these genes with the gene Pd*mfs2* knockout indicated some compensatory effects in the potential to sustain prochloraz resistance of the *mfs2*-defective *P. digitatum* strain.

### 3.6. Drug Pump Protein-Encoding Gene Expression Profiles

In the present study, a series of homologous genes encoding the three main types of drug pump proteins, including MFS-, ABC- and MATE-encoding isogenes, were selected to investigate their fold changes in the four comparative groups. The MFS-, ABC- and MATE proteins encoded by these isogenes were subjected to multiple amino acid-sequence alignments, respectively, using Clustal_X2 software and online tool ENDscript server (version 3.0). As shown in Appendix A, there were significantly high homologs in the primary structure and the classical motifs between all the 33 MFS isoforms (i.e., MFS1~MFS33). The similar results were obtained in the multiple sequence alignments of ABC (i.e., ABC1~ABC8) and MATE (MATE1~MATE3) proteins, respectively (Appendix A).

As summarized in Table 8, drug-pump gene *MFS2* was up-regulated in the wild-type *P. digitatum* strain with prochloraz induction, but such an up-regulation was not observed in the *mfs2*-defective strain due to the knockout of the MFS2-encoding gene. In the comparison Pd-wt-(I/NI), besides the MFS2-encoding gene, the other two MFS isogenes were up-regulated after prochloraz treatment, i.e., *MFS21* (PDIP_55680) and *MFS22* (PDIP_19590). These two MFS-encoding genes were not up-regulated in the *mfs2*-defective *P. digitatum* strain with prochloraz treatment. In the absence of prochloraz, the knockout of Pd*mfs2* alone also led to the down-regulation of multiple *MFS*isogenes, including *MFS1* (PDIP_66230), *MFS3* (PDIP_34090), *MFS4* (PDIP_53210), *MFS5* (PDIP_21030), *MFS8* (PDIP_77890), *MFS9* (PDIP_77880), *MFS12* (PDIP_57820), *MFS15* (PDIP_18570), and *MFS18* (PDIP_11120). These MFS-encoding genes were also down-regulated in the comparison I-(Pd-d/Pd-wt) with similar fold changes to those of the comparison NI-(Pd-d/Pd-wt). Such lower transcript abundances of the above nine MFS-encoding genes might cause lower prochloraz resistance for the *mfs2*-defective *P. digitatum* strain. However, the nine MFS-encoding genes down-regulated in the two comparisons (i.e., I-(Pd-d/Pd-wt) and NI-(Pd-d/Pd-wt)) cannot be induced in the *mfs2*-defective *P. digitatum* strain after prochloraz treatment. Such transcriptional evidence also indicated these nine MFS-encoding genes might play roles in sustain the essential baselines of prochloraz resistance for both wild-type and *mfs2*-defective *P. digitatum* strains.

On the other hand, in the absence of prochloraz, the knockout of Pd*mfs2* alone also led to the up-regulation of multiple MFS isogenes, including *MFS23* (PDIP_36610), *MFS26* (PDIP_64100), *MFS27* (PDIP_55370), *MFS31* (PDIP_19850), *MFS33* (PDIP_55020) (Table 8). All these five MFS-encoding genes did not show up-regulation in the comparison Pd-d-(I/NI); however, three of them (i.e., *MFS23*, *MFS26* and *MFS27*) showed up-regulation in the comparison I-(Pd-d/Pd-wt). These three MFS-encoding genes might exert some compensatory effects to sustain the prochloraz resistance of the *mfs2*-defective *P. digitatum* strain. Interestingly, in the comparison I-(Pd-d/Pd-wt), the amount of down-regulated MFS isogenes was obviously higher than that of up-regulated MFS isogenes. Such a profile might indicate that the compensatory effects by several MFS isogenes in potential could not compensate in full for the loss of Pd*mfs2* gene that induced a simultaneous down-regulation of most of the MFS-encoding genes.

In addition to the MFS-encoding genes, another class of drug pump protein-encoding genes (i.e., ABC-encoding genes) exhibited similar changing profiles (Table 8), as compared to MFS isogenes. After prochloraz treatment, the only one ABC-encoding gene, i.e., *ABC2* (PDIP_58890), was up-regulated in the wild-type *P. digitatum* strain after prochloraz treatment, but not up-regulated in the *mfs2*-defective *P. digitatum* strain. Considering there was not any other ABC-encoding gene up-regulated in the wild-type *P. digitatum* strain after prochloraz treatment, the ABC2-encoding gene might be the essential contributor to developing prochloraz resistance. On the other hand, the knockout of Pd*mfs2* led to the up-regulation of multiple ABC isogenes in the absence of prochloraz, including *ABC3* (PDIP_13640), *ABC4* (PDIP_19230), *ABC5* (PDIP_78490), *ABC6* (PDIP_37050), *ABC7* (PDIP_37060), and *ABC8* (PDIP_57360). Among them, the genes encoding ABC3, ABC4 and ABC5 were simultaneously up-regulated in the comparison I-(Pd-d/Pd-wt). These three ABC-encoding genes might exert some compensatory effects to sustain prochloraz resistance of the *mfs2*-defective *P. digitatum* strain. Regarding the third class of drug pump protein-encoding genes (i.e., MATE-encoding genes), all the three MATE isogenes were up-regulated in the wild-type *P. digitatum* strain after prochloraz treatment, including *MATE1* (PDIP_56750), *MATE2* (PDIP_40930), and *MATE3* (PDIP_05620) (Table 8). However, such up-regulation was not observed in the comparison Pd-d-(I/NI). That is to say, the three MATE-encoding genes were not up-regulated in the *mfs2*-defective *P. digitatum* strain after prochloraz treatment. Not similar to the changing profiles of MFS- and ABC-encoding genes, none of the present MATE isogenes were induced by the Pd*mfs2* gene knockout at no prochloraz treatment, and accordingly, the three MATE-encoding genes were simultaneously down-regulated in the comparison I-(Pd-d/Pd-wt). These lines of evidence indicated that the decreased prochloraz resistance of the *mfs2*-defective *P. digitatum* strain might be in part due to the loss of Pd*mfs2* gene that led to the simultaneous down-regulation of all three MATE isogenes.

### 3.7. qPCR Validation of DEGs

The results of qPCR validation for the selected DEGs are summarized in Table 9, including MFS-encoding genes, ABC-encoding genes, MATE-encoding genes, and multiple metabolism-relating and stress-responsive protein-encoding genes. In general, the transcript abundance change profiles of all DEGs in the present four comparative groups, obtained using qPCR, were well correlated with those obtained by RNA-seq analysis. At prochloraz induction, with comparison to no prochloraz induction, some of the up-regulated MFS-, ABC-, and MATE-encoding genes in the wild-type *P. digitatum* strain did exhibit no up-regulation or lower folds of up-regulation in the *mfs2*-defective *P. digitatum* strain, including *MFS6*, *MFS7*, *MFS10*, *MFS16*, *MFS21*, *MFS22*, *ABC2*, *ABC8*, *MATE1*, *MATE2*, and *MATE3*. Similar changing profiles were also found in those of multiple metabolism-relating and stress-responsive protein-encoding genes.

## 4. Discussion

Pd*mfs2* has been reported as an essential gene to develop high prochloraz resistance of the *P. digitatum* strain, as the knockout of this drug-pump protein-encoding gene led to a significantly lower prochloraz resistance [8]. The underlying mechanisms need further studies regarding how the Pd*mfs2* regulates fungal prochloraz resistance.

According to studies in the past years, more and more evidence has emerged to support comprehensive metabolism backgrounds underlying fungicide resistance. Such backgrounds include various ERG-encoding genes in fungal ergosterol biosynthesis pathways [36,41,42], acetyl-CoA carboxylase (ACCase)-encoding genes in the lipid and fatty acid oxidation pathways [45,46,47], reactive oxygen species (ROS)-metabolizing enzyme-encoding genes in the cell wall maintenance and oxidative-stress-responsive processes [50,51], mitochondrial respiratory chain protein-encoding genes in the cellular energy metabolisms [53,54,55], ubiquitin-encoding genes in the post-translational modification processes [57,58,59], and protein kinase-encoding genes involved in mitogen-activated signal transductions [55,60,61]. In the present study, RNA-seq analysis revealed that Pd*mfs2* knockout led to the down-regulation of genes involved in peroxisome (ko04146) and oxidative phosphorylation (ko00190) at no prochloraz treatment (Table 1 and Table 2, and Figure 1 and Figure 2). Specially, the down-regulation of lipid metabolism-relating genes in the comparison NI-(Pd-d/Pd-wt), including fatty acyl-CoA oxidase-encoding gene and carnitine acetyl transferase-encoding gene (Table 2), was also reported in *Pisolithusmicrocarpus* and *Beauveria bassiana* [76,77]. And the down-regulation of energy metabolism-relating genes in the comparison NI-(Pd-d/Pd-wt), including cytochrome b-encoding gene and ATP synthase-encoding gene (Table 2), was also reported in *Botrytis cinerea* and *Corynespora cassiicola* [55,78]. Thus, even in the absence of fungicide, there has been an association of Pd*mfs2* with multiple metabolisms required in the fungi adaptation to their growth environments, and such an adaptation might be one aspect of the physiological basis to develop fungicide resistance.

In the present RNA-seq analysis, the genes involved in oxidative phosphorylation, steroid biosynthesis, biosynthesis of fatty acids including unsaturated fatty acids, ubiquinone biosynthesis, and ribosome processes were all up-regulated in the wild-type *P. digitatum* strain after prochloraz treatment (Table 3 and Figure 3). Such a simultaneous up-regulation in the prochloraz induction suggested that the prochloraz resistance did require these multiple metabolism backgrounds. Similar requirements were observed in the prochloraz-treated *mfs2*-defective *P. digitatum* strain (Table 4 and Figure 3). However, after prochloraz treatment, the up-regulated KEGG classes in the *mfs2*-defective *P. digitatum* strain were obviously less than those in the wild-type *P. digitatum* strain (Table 3 and Table 4). Actually, many up-regulated DEGs in the comparison Pd-wt-(I/NI), relating to ergosterol biosynthesis, lipid and fatty acid oxidation, oxidative-stress-responsive processes, and cellular energy metabolisms, were not identified as DEGs in the comparison Pd-d-(I/NI) (Table 5 and Figure 4, Figure 5 and Figure 6). For example, neither NADH dehydrogenase-encoding genes, nor cytochrome b-encoding gene, cytochrome c oxidase-encoding gene, ATP synthase-encoding gene, Erg-encoding genes (i.e., *Erg1*, *Erg3*, and *Erg25*), or ribosomal protein-encoding genes were up-regulated in the *mfs2*-defective *P. digitatum* strain (Table 5). The present results indicated the importance of Pd*mfs2* in developing prochloraz resistance. That is to say, the Pd*mfs2* might be functionally associated with multiple metabolism-relating genes, and they are cooperatively expressed to confer *P. digitatum* prochloraz resistance through specific mechanisms that need further study.

On the other hand, the down-regulated DEGs of the comparison I-(Pd-d/Pd-wt) were mainly enriched into ‘ribosome’ and ‘oxidative phosphorylatio’ KEGG pathways (Table 6 and Figure 7), including ribosomal protein-encoding genes, cytochrome c oxidase-encoding gene, and ATP synthase-encoding gene. Similar results were obtained in the KOG classification (Table 7 and Figure 8). At the prochloraz treatment, the transcriptional abundances of these genes in the *mfs2*-defective *P. digitatum* strain were all lower than those of the wild-type *P. digitatum* strain. The results suggested (1) the important functions of these genes to sustain fungal prochloraz resistance, and (2) the potential association of these genes with Pd*mfs2*. In contrast, some genes were up-regulated in the comparison I-(Pd-d/Pd-wt) (Table 6 and Table 7). These genes and the relating metabolic pathways might in part compensate for the Pd*mfs2* gene-deletion, sustaining the prochloraz resistance baseline of the *mfs2*-defective *P. digitatum* strain.

Multiple iso-genes encoding MFS, ABC, and MATE transporters have been identified in the fungal genomics and RNA-seq studies [32,33]. Simultaneous over-expression of MFS and ABC genes has been found in the highly prochloraz-resistant fungus [9,26,27,28]. Similar results were obtained in the present study, as multiple MFS-, ABC-, and MATE-encoding genes were up-regulated in the wild-type *P. digitatum* strain, whereas most of the putative drug-pump genes were not up-regulated in the *mfs2*-defective *P. digitatum* strain (Table 8 and Table 9). This correlation of Pd*mfs2* with other drug-pump genes suggested the importance of Pd*mfs2* in the drug-pump-induced prochloraz resistance. On the other hand, the over-expression of specific MFS- and ABC-encoding genes cannot compensate for the loss of Pd*mfs2*, which further verifies the important role of Pd*mfs2* in developing the prochloraz resistance.

## 5. Conclusions

The present study provided some transcriptome evidence regarding the important role of drug-efflux pump protein gene *mfs2* in *P. digitatum* prochloraz resistance. The knockout of *mfs2* led to the simultaneous down-regulation of other drug-efflux pump protein gene expression, and led to the simultaneous down-regulation of cellular metabolism-related gene expression, including ribosome biosynthesis-related genes, oxidative phosphorylation genes, steroid biosynthesis-related genes, fatty acid biosynthesis-related genes, and carbon- and nitrogen-metabolism-related genes. These results indicated a more comprehensive background, regarding a crosslink between various drug-efflux pump proteins and between multiple cellular metabolisms, which might be associated with *mfs2*-introduced prochloraz resistance in the *P. digitatum* strain (PdF6).

## Figures and Tables

**Figure 1 microorganisms-12-00888-f001:**
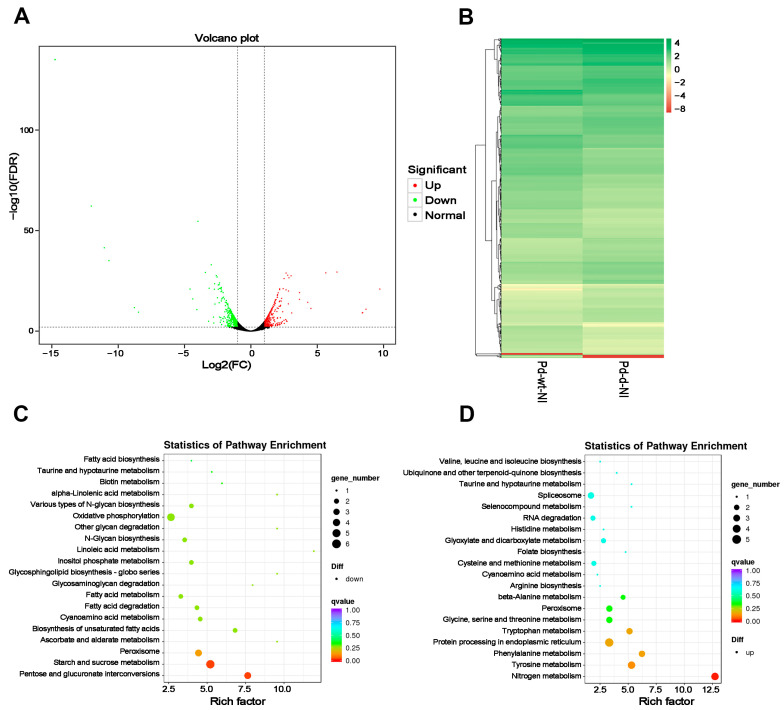
DEG analysis between the wild-type and *mfs2*-deleted *P. digitatum* strains at no prochloraz induction. (**A**) Volcano analysis of all DEGs; (**B**) heatmap analysis of all DEGs; (**C**) KEGG enrichment of down-regulated DEGs; (**D**) KEGG enrichment of up-regulated DEGs. Pd-wt-NI indicated the wild-type *P. digitatum* strain at no prochloraz induction. Pd-d-NI indicated the *mfs2*-deleted *P. digitatum* strain at no prochloraz induction. In the panel (**A**), the two dashed lines at vertical axis indicated the cut-off values to define DEGs in terms of log2(FC), i.e., “−1” for the left line and “1” for the right; the one dashed line at horizontal axis indicated the cut-off value (i.e., “2”) to define DEGs in terms of −log10(FDR).

**Figure 2 microorganisms-12-00888-f002:**
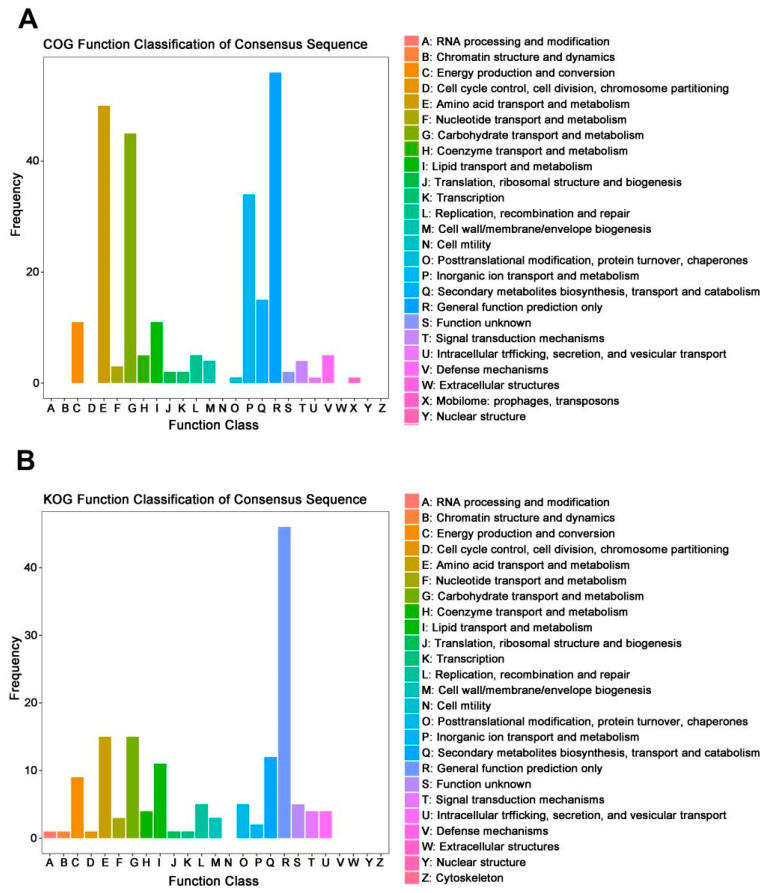
COG and KOG function classification of down-regulated DEGs between the wild-type and *mfs2*-deleted *P. digitatum* strains at no prochloraz induction. (**A**) COG classification; (**B**) KOG classification.

**Figure 3 microorganisms-12-00888-f003:**
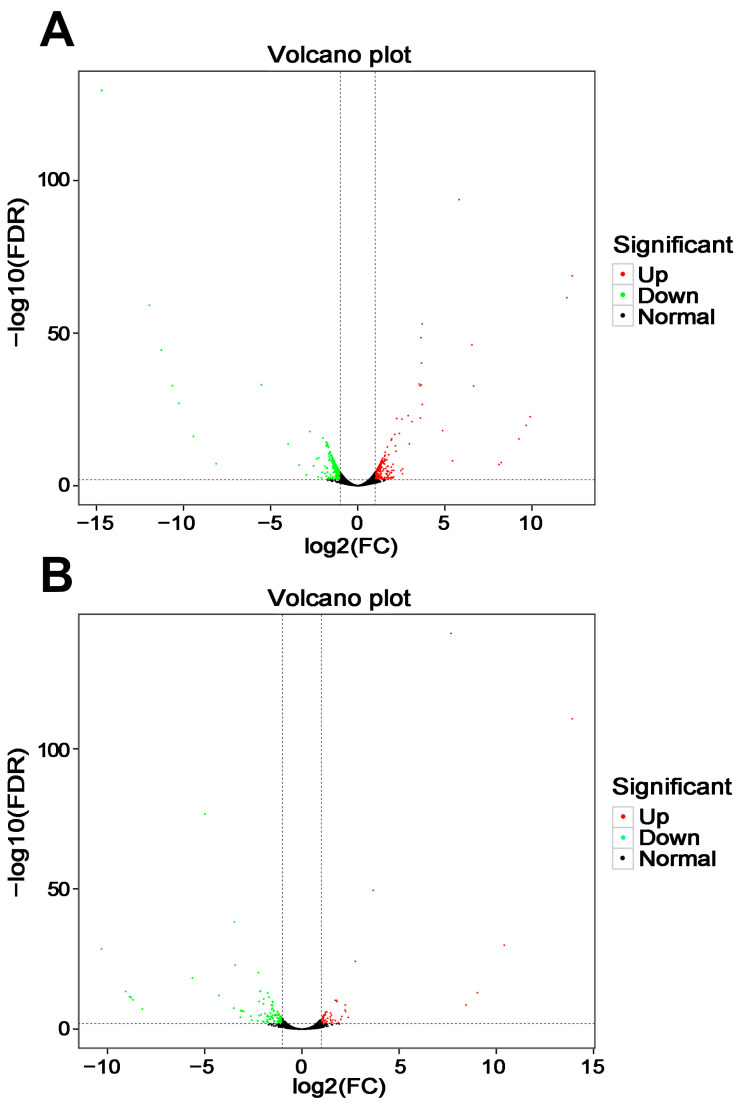
Volcano analysis of DEGs in the wild-type and *mfs2*-deleted *P. digitatum* strains with prochloraz induction. (**A**) Volcano analysis of DEGs in the wild-type *P. digitatum* strain with prochloraz induction (Pd-wt-I) in comparison to that with no prochloraz induction (Pd-wt-NI); (**B**) volcano analysis of DEGs in the *mfs2*-deleted *P. digitatum* strain with prochloraz induction (Pd-d-I) in comparison to that with no prochloraz induction (Pd-d-NI). The two dashed lines at vertical axis indicated the cut-off values to define DEGs in terms of log2(FC), i.e., “−1” for the left line and “1” for the right; the one dashed line at horizontal axis indicated the cut-off value (i.e., “2”) to define DEGs in terms of −log10(FDR).

**Figure 4 microorganisms-12-00888-f004:**
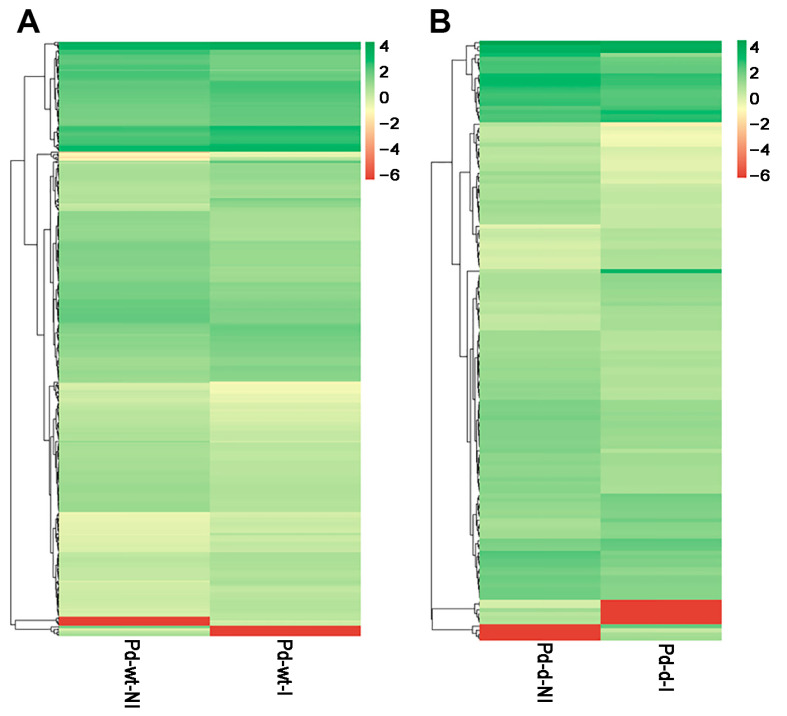
Heatmap clustering of DEGs in the wild-type and *mfs*2-deleted *P. digitatum* strains with prochloraz induction. (**A**) Heatmap clustering of DEGs in the wild-type *P. digitatum* strain with prochloraz induction (Pd-wt-I) in comparison to that with no prochloraz induction (Pd-wt-NI); (**B**) heatmap clustering of DEGs in the *mfs*2-deleted *P. digitatum* strain with prochloraz induction (Pd-d-I) in comparison to that with no prochloraz induction (Pd-d-NI).

**Figure 5 microorganisms-12-00888-f005:**
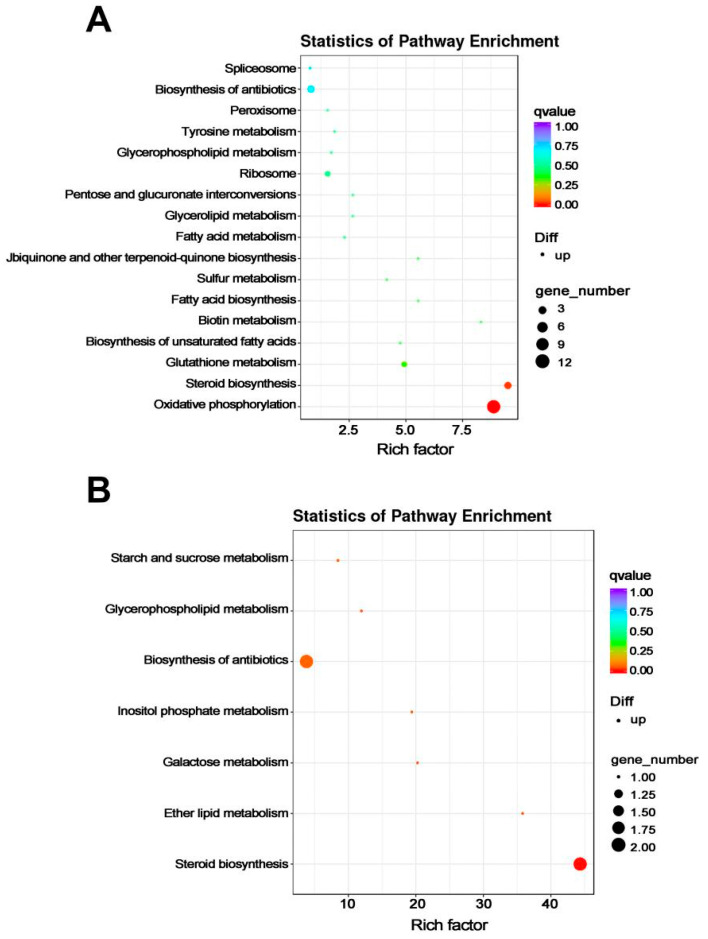
KEGG enrichment of up-regulated DEGs in the wild-type and *mfs*2-deleted *P. digitatum* strains with prochloraz induction. (**A**) KEGG enrichment of up-regulated DEGs in the wild-type *P. digitatum* strain with prochloraz induction (Pd-wt-I) in comparison to that with no prochloraz induction (Pd-wt-NI); (**B**) KEGG enrichment of up-regulated DEGs in the *mfs*2-deleted *P. digitatum* strain with prochloraz induction (Pd-d-I) in comparison to that with no prochloraz induction (Pd-d-NI).

**Figure 6 microorganisms-12-00888-f006:**
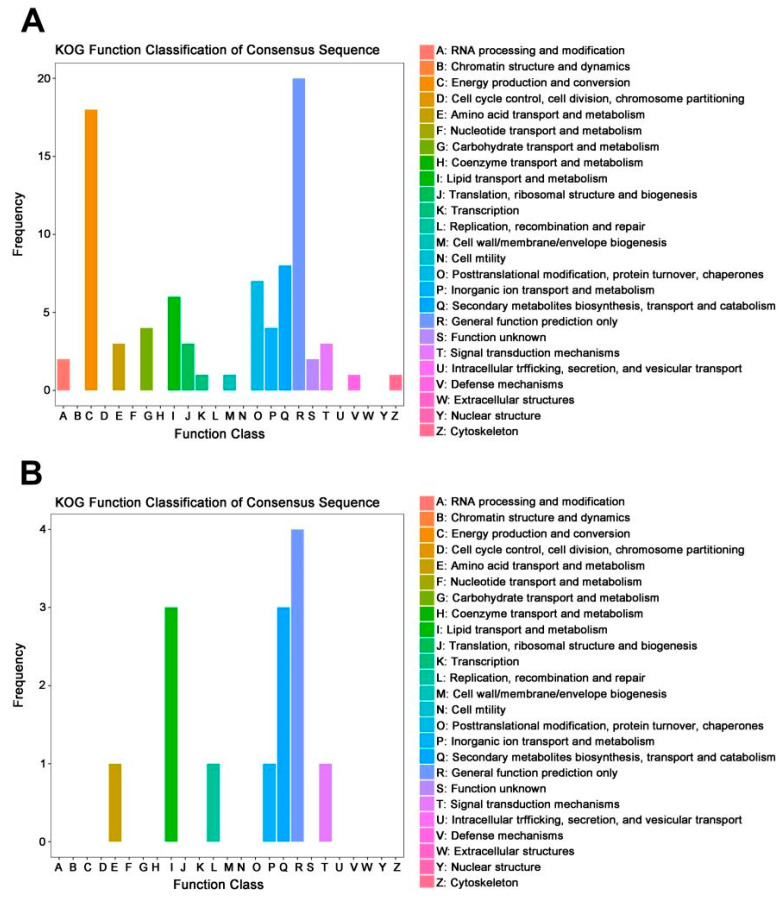
KOG function classification of up-regulated DEGs in the wild-type and *mfs2*-deleted *P. digitatum* strains with prochloraz induction. (**A**) KOG function classification of up-regulated DEGs in the wild-type *P. digitatum* strain with prochloraz induction (Pd-wt-I) in comparison to that with no prochloraz induction (Pd-wt-NI); (**B**) KOG function classification of up-regulated DEGs in the *mfs2*-deleted *P. digitatum* strain with prochloraz induction (Pd-d-I) in comparison to that with no prochloraz induction (Pd-d-NI).

**Figure 7 microorganisms-12-00888-f007:**
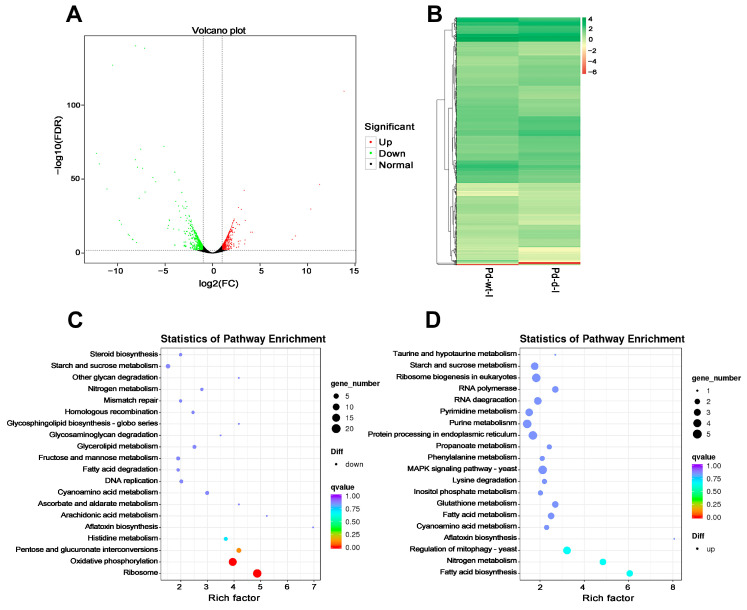
DEG analysis between the wild-type and *mfs2*-deleted *P. digitatum* strains at prochloraz induction. (**A**) Volcano analysis of all DEGs; (**B**) heatmap analysis of all DEGs; (**C**) KEGG enrichment of down-regulated DEGs; (**D**) KEGG enrichment of up-regulated DEGs. Pd-wt-I indicated the wild-type *P. digitatum* strain at prochloraz induction. Pd-d-I indicated the mfs2-deleted *P. digitatum* strain at prochloraz induction. In the panel A, the two dashed lines at vertical axis indicated the cut-off values to define DEGs in terms of log2(FC), i.e., “−1” for the left line and “1” for the right; the one dashed line at horizontal axis indicated the cut-off value (i.e., “2”) to define DEGs in terms of −log10(FDR).

**Figure 8 microorganisms-12-00888-f008:**
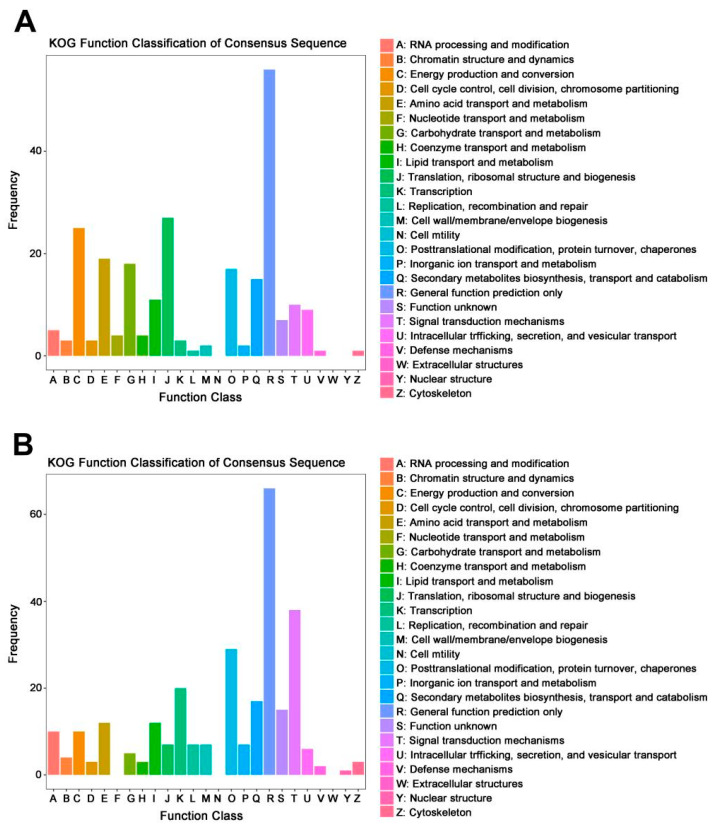
KOG function classification of DEGs between the wild-type and mfs2-deleted *P. digitatum* strains at prochloraz induction. (**A**) KOG function classification of down-regulated DEGs in the *mfs2*-deleted *P. digitatum* strain (Pd-d-I) in comparison to wild-type *P. digitatum* strain (Pd-wt-I) at prochloraz induction; (**B**) KOG function classification of up-regulated DEGs in the *mfs2*-deleted *P. digitatum* strain (Pd-d-I) in comparison to wild-type *P. digitatum* strain (Pd-wt-I) at prochloraz induction.

**Table 1 microorganisms-12-00888-t001:** KEGG-enriched DEGs between the wild-type and *mfs2*-deleted *P. digitatum* strains at no prochloraz induction.

KEGG (ID)	Annotated Function of DEG	Regulated	Log2FC	FDR Value
Pentose and glucuronate interconversions (ko00040)	Dihydrodipicolinate synthetase	Down	−1.54	1.66 × 10^−10^
	Exopolygalacturonase	Down	−1.07	2.31 × 10^−4^
Starch and sucrose metabolism (ko00500)	Exo-β-1,3-glucanase	Down	−1.25	3.74 × 10^−5^
	α-L-Rhamnosidase	Down	−1.46	8.31 × 10^−8^
	β-Glucosidase	Down	−1.12	2.35 × 10^−5^
	Exopolygalacturonase	Down	−1.07	2.31 × 10^−4^
Peroxisome (ko04146)	Peroxin	Down	−1.23	9.75 × 10^−7^
	Fatty acyl-CoA oxidase	Down	−1.14	1.40 × 10^−6^
	Carnitine acetyl transferase	Down	−1.05	1.07 × 10^−5^
Oxidative phosphorylation (ko00190)	NADH dehydrogenase	Down	−1.93	3.17 × 10-^3^
	Cytochrome b	Down	−2.27	1.52 × 10^−7^
	ATP synthase	Down	−1.45	5.26 × 10^−3^
Nitrogen metabolism (ko00910)	NAD^+^-dependent glutamate dehydrogenase	Up	1.04	1.10 × 10^−5^
	Nitrite reductase	Up	1.32	1.55 × 10^−8^
	Nitrate reductase	Up	1.19	1.27 × 10^−6^
	Nitrilase	Up	1.72	1.19 × 10^−6^
Tyrosine metabolism (ko00350)	Maleylacetoacetate isomerase	Up	1.47	2.30 × 10^−6^
	4-Hydroxyphenylpyruvate dioxygenase	Up	1.62	2.28 × 10^−12^
	Amine oxidase	Up	1.66	3.46 × 10^−13^
	Aldehyde dehydrogenase	Up	1.02	1.69 × 10^−5^
Phenylalanine metabolism (ko00360)	4-Hydroxyphenylpyruvate dioxygenase	Up	1.62	2.28 × 10^−12^
	Amine oxidase	Up	1.66	3.46 × 10^−13^
	Aldehyde dehydrogenase	Up	1.02	1.69 × 10^−5^
Tryptophan metabolism (ko00380)	Nitrilase	Up	1.17	1.19 × 10^−6^
	Indoleamine/pyrrole 2,3-dioxygenase	Up	1.07	8.21 × 10^−6^
	Catalase	Up	1.13	1.65 × 10^−5^

**Table 2 microorganisms-12-00888-t002:** DEGs in the COG and KOG function classification of down-regulated DEGs between the wild-type and *mfs2*-deleted *P. digitatum* strains at no prochloraz induction.

Annotated Function of DEG	Database	Class Name	ID
Dihydrodipicolinate	COG	Amino acid transport and metabolism	E
	COG	Cell wall/membrane/envelope biogenesis	M
Exo-β-1,3-glucanase	COG	Carbohydrate transport and metabolism	G
α-L-Rhamnosidase	COG	Carbohydrate transport and metabolism	G
	KOG	Carbohydrate transport and metabolism	G
β-Glucosidase	COG	Carbohydrate transport and metabolism	G
Peroxin	COG	General function prediction only	R
	KOG	Intracellular trafficking, secretion, and vesicular transport	U
Fatty acyl-CoA oxidase	COG	Lipid transport and metabolism	I
	KOG	Lipid transport and metabolism	I
Carnitine acetyl transferase	KOG	Lipid transport and metabolism	I
NADH dehydrogenase	COG	Energy production and conversion	C
	KOG	Energy production and conversion	C
Cytochrome b	COG	Energy production and conversion	C
	KOG	Energy production and conversion	C
ATP synthase	COG	Energy production and conversion	C
	KOG	Energy production and conversion	C

**Table 3 microorganisms-12-00888-t003:** Up-regulated DEGs involved in KEGG enrichment and classification in the wild-type *P. digitatum* strain with prochloraz induction (i.e., Pd-wt-(I/NI)).

KEGG (ID)	Annotated Function of DEG	Log2FC	FDR
Oxidative phosphorylation (ko00190)	Cytochrome c oxidase	2.54	1.54 × 10^−22^
	NADH dehydrogenase	3.65	8.77 × 10^−34^
	Cytochrome b	2.39	6.76 × 10^−18^
	ATP synthase	3.53	4.74 × 10^−34^
Steroid biosynthesis (ko00100)	Erg24	1.30	5.65 × 10^−8^
	Erg1	1.35	1.17 × 10^−8^
	Erg25	1.43	9.29 × 10^−10^
Glutathione metabolism (ko00480)	Glutathione S-transferase	1.13	1.03 × 10^−3^
Biosynthesis of unsaturated fatty acids (ko01040)	1,3,6,8-Tetrahydroxynaphthalene reductase	1.08	9.74 × 10^−5^
Biotin metabolism (ko00780)	1,3,6,8-Tetrahydroxynaphthalene reductase	1.08	9.74 × 10^−5^
Fatty acid biosynthesis (ko00061)	1,3,6,8-Tetrahydroxynaphthalene reductase	1.08	9.74 × 10^−5^
Ubiquinone and other terpenoid-quinone biosynthesis (ko00130)	NADH-quinone oxidoreductase	1.43	8.80 × 10^−10^
Fatty acid metabolism (ko01212)	1,3,6,8-Tetrahydroxynaphthalene reductase	1.08	9.74 × 10^−5^
Glycerolipid metabolism (ko00561)	Glycerol kinase	1.62	1.24 × 10^−4^
Pentose and glucuronate interconversions (ko00040)	Pectate lyase	1.51	1.02 × 10^−8^
Ribosome (ko03010)	40S Ribosomal protein	1.01	3.49 × 10^−5^
	Ribosomal protein	1.01	3.00 × 10^−5^
Glycerophospholipid metabolism (ko00564)	Phosphatidylserine decarboxylase	1.14	2.23 × 10^−6^
Tyrosine metabolism (ko00350)	Maleylacetoacetate isomerase	1.10	1.42 × 10^−3^
Peroxisome (ko04146)	Superoxide dismutase	1.25	1.90 × 10^−7^
Biosynthesis of antibiotics (ko01130)	Erg25	1.43	9.29 × 10^−10^
	Erg24	1.30	5.65 × 10^−8^
	Erg1	1.35	1.17 × 10^−8^
Spliceosome (ko03040)	Pre-mRNA-splicing factor	1.12	3.43 × 10^−6^

**Table 4 microorganisms-12-00888-t004:** Up-regulated DEGs involved in KEGG enrichment and classification in the *mfs*2-deleted P. digitatum strain with prochloraz induction in comparison to that without prochloraz induction (i.e., Pd-d-(I/NI)).

KEGG (ID)	Annotated Function of DEG	Log2FC	FDR
Steroid biosynthesis (ko00100)	Erg3	1.11	1.09 × 10^−5^
	Erg25	1.12	9.08 × 10^−6^
Ether lipid metabolism (ko00565)	Phospholipase C	1.46	3.07 × 10^−4^
Galactose metabolism (ko00052)	Extracellular invertase	1.88	4.25 × 10^−3^
Inositol phosphate metabolism (ko00562)	Phospholipase C	1.46	3.07 × 10^−4^
Biosynthesis of antibiotics (ko01130)	Erg3	1.11	1.09 × 10^−5^
	Erg25	1.12	9.08 × 10^−6^
Glycerophospholipid metabolism (ko00564)	Phospholipase C	1.46	3.07 × 10^−4^
Starch and sucrose metabolism (ko00500)	Extracellular invertase	1.88	4.25 × 10^−3^

**Table 5 microorganisms-12-00888-t005:** KOG function classification of up-regulated DEGs in the Pd-wt-(I/NI) and Pd-d-(I/NI).

Function Class	Annotated Function of DEG	Pd-wt-(I/NI)		Pd-d-(I/NI)	
		Log2FC	FDR	Log2FC	FDR
A	Zinc knuckle transcription factor	1.08	2.47 × 10^−4^	/	/
	RNA helicase	2.57	1.19 × 10^−4^	/	/
C	RNA helicase	2.57	1.19 × 10^−4^	/	/
	NADH dehydrogenase subunit 4	2.90	1.11 × 10^−23^	/	/
	NADH dehydrogenase subunit 1	2.02	8.60 × 10^−8^	/	/
	Hypothetical protein	1.15	1.34 × 10^−3^	/	/
	Glyoxylate reductase	1.14	5.55 × 10^−4^	/	/
	Hypothetical protein	1.14	8.06 × 10^−6^	/	/
	Cytochrome b	2.39	6.76 × 10^−18^	/	/
	Hypothetical protein	1.12	3.43 × 10^−6^	/	/
	Cytochrome c oxidase	2.54	1.54 × 10^−22^	/	/
	ATP synthase subunit 9	2.47	9.84 × 10^−6^	/	/
	ATP synthase subunit 6	3.53	4.74 × 10^−34^	/	/
	Oxaloacetate hydrolase	3.70	2.35 × 10^−27^	/	/
	Hypothetical protein	1.21	6.71 × 10^−4^	/	/
	FMN dependent dehydrogenase	1.16	6.23 × 10^−6^	/	/
E	Nitrilase	1.06	1.30 × 10^−5^	/	/
	Amino acid permease	1.17	4.03 × 10^−6^	/	/
G	Glycerol kinase	1.62	1.24 × 10^−4^	/	/
	MFS	1.29	1.42 × 10^−6^	/	/
	Aquaporin	1.25	1.51 × 10^−7^	/	/
I	NRPS-like enzyme	1.28	3.99 × 10^−4^	/	/
	Phosphatidylserine decarboxylase	1.14	2.23 × 10^−6^	/	/
	Erg3	/	/	1.11	1.09 × 10^−5^
	Erg25	1.43	9.29 × 10^−10^	1.12	9.08 × 10^−6^
	Epoxide hydrolase	1.08	1.85 × 10^−5^	/	/
	C-14 sterol reductase	1.30	5.65 × 10^−8^	/	/
J	Ribosomal protein	1.01	3.00 × 10^−5^	/	/
	40S Ribosomal protein	1.01	3.49 × 10^−5^	/	/
O	Protein-L-isoaspartate O-methyltransferase	1.22	1.22 × 10^−6^	/	/
	Thioredoxin	1.00	4.10 × 10^−5^	/	/
	Glutathione S-transferase	1.13	1.04 × 10^−3^	/	/
	Maleylacetoacetate isomerase	1.10	1.42 × 10^−3^	/	/
P	Metabolite transport protein GIT1	1.74	1.93 × 10^−3^	/	/
	Superoxide dismutase	1.24	1.90 × 10^−7^	/	/
	Plasma membrane low affinity zinc ion transporter	/	/	1.21	8.06 × 10^−3^
Q	Isopenicillin N synthase	1.05	8.48 × 10^−5^	/	/
	ABC	1.52	3.91 × 10^−3^	/	/
	Phenyloxazoline synthase	1.08	3.42 × 10^−3^	/	/
	Alcohol dehydrogenase	1.25	1.21 × 10^−6^	/	/
	Flavin-binding monooxygenase-like protein	/	/	1.10	1.71 × 10^−4^
R	NADH-quinone oxidoreductase	1.43	8.80 × 10^−10^	/	/
	Aldehyde reductase	1.16	6.37 × 10^−3^	/	/
	Short chain dehydrogenase/reductase	1.16	1.74 × 10^−5^	/	/
	Isopenicillin N synthase	1.05	8.48 × 10^−5^	/	/
	Dienelactone hydrolase	1.05	2.81 × 10^−4^	/	/
	Short chain dehydrogenase/reductase	1.04	2.65 × 10^−5^	/	/
	1,3,6,8-Tetrahydroxynaphthalene reductase	1.08	9.74 × 10^−5^	/	/
	Pre-mRNA-splicing factor	1.12	3.43 × 10^−6^	/	/
	Carbonyl reductase	1.57	7.64 × 10^−11^	/	/
T	Erg1	1.35	1.17 × 10^−8^	/	/
	C-14 sterol reductase	1.30	5.65 × 10^−8^	/	/
Z	Profilin	1.03	3.34 × 10^−4^	/	/

**Table 6 microorganisms-12-00888-t006:** KEGG-enriched DEGs between the wild-type and *mfs2*-deleted *P. digitatum* strains at prochloraz induction.

KEGG (ID)	Annotated Function of DEG	Regulated	Log2FC	FDR
Ribosome (ko03010)	60S Ribosomal protein	Down	−1.28	1.84 × 10^−8^
	40S Ribosomal protein	Down	−1.36	2.11 × 10^−9^
	Ribosomal protein	Down	−1.43	2.69 × 10^−10^
	60S Acidic ribosomal phosphoprotein	Down	−1.02	9.34 × 10^−6^
Oxidative phosphorylation (ko00190)	Cytochrome c oxidase	Down	−7.57	9.35 × 10^−141^
	ATP synthase	Down	−8.78	1.13 × 10^−12^
	NADH dehydrogenase	Down	−11.12	5.82 × 10^−44^
	Cytochrome b	Down	−6.28	1.02 × 10^−51^
	ATPase proteolipid	Down	−1.04	1.03 × 10^−5^
	NADH-ubiquinone oxidoreductase	Down	−1.33	3.66 × 10^−8^
Pentose and glucuronate interconversions (ko00040)	Mandelate racemase/muconate lactonizing enzyme	Down	−1.09	2.49 × 10^−6^
	Exopolygalacturonase	Down	−1.60	4.24 × 10^−8^
Fatty acid biosynthesis (ko00061)	Fatty acid synthase β subunit	Up	1.38	1.23 × 10^−9^
	Fatty acid synthase α subunit	Up	1.25	3.66 × 10^−8^
	Acetyl-CoA carboxylase	Up	1.73	1.67 × 10^−14^
Nitrogen metabolism (ko00910)	NAD^+^-dependent glutamate dehydrogenase	Up	2.04	1.46 × 10^−19^
	Nitrite reductase	Up	1.29	2.42 × 10^−8^
	Nitrilase	Up	1.26	6.59 × 10^−8^
Regulation of mitophagy—yeast (ko04139)	Transcription factor	Up	1.07	3.42 × 10^−6^
	MAP kinase kinasekinase	Up	1.34	5.63 × 10^−9^
	Ubiquitin carboxyl-terminal hydrolase	Up	1.14	1.03 × 10^−6^

**Table 7 microorganisms-12-00888-t007:** KOG function classification of DEGs between the wild-type and *mfs2*-deleted *P. digitatum* strains at prochloraz induction.

Annotated Function of DEG	Regulated	Class Name	Class ID
60S Ribosomal protein	Down	Translation, ribosomal structure and biogenesis	J
40S Ribosomal protein	Down	Translation, ribosomal structure and biogenesis	J
Ribosomal protein	Down	Translation, ribosomal structure and biogenesis	J
60S Acidic ribosomal phosphoprotein	Down	Translation, ribosomal structure and biogenesis	J
Cytochrome c oxidase	Down	Energy production and conversion	C
ATP synthase	Down	Energy production and conversion	C
NADH dehydrogenase	Down	Energy production and conversion	C
Cytochrome b	Down	Energy production and conversion	C
ATPase proteolipid	Down	Energy production and conversion	C
NADH-ubiquinone oxidoreductase	Down	Energy production and conversion	C
Acetyl-CoA carboxylase	Up	Lipid transport and metabolism	I
NAD^+^-dependent glutamate dehydrogenase	Up	Amino acid transport and metabolism	E
Nitrite reductase	Up	General function prediction only	R
Nitrilase	Up	Amino acid transport and metabolism	E
Transcription factor	Up	Chromatin structure and dynamics	B
MAP kinase kinasekinase	Up	Signal transduction mechanisms	T
Ubiquitin carboxyl-terminal hydrolase	Up	Post-translational modification, protein turnover, chaperones	O

**Table 8 microorganisms-12-00888-t008:** Changing fold (log2FC) of drug pump protein homologous genes in four comparative groups.

Gene Name	Changing Fold (log2FC) of the Gene Transcription Abundance in the Below Groups in the Present Comparative Analysis			
	Pd-wt-(I/NI)	Pd-d-(I/NI)	I-(Pd-d/Pd-wt)	NI-(Pd-d/Pd-wt)
*MFS1* (PDIP_66230)	/	/	−2.28	−2.46
*MFS2* (PDIP_88410)	1.18	0	0	0
*MFS3* (PDIP_34090)	/	/	−1.11	−1.40
*MFS4* (PDIP_53210)	/	/	−1.93	−1.33
*MFS5* (PDIP_21030)	/	/	−1.33	−1.31
*MFS6* (PDIP_86550)	/	/	−1.20	/
*MFS7* (PDIP_02580)	/	/	−2.30	/
*MFS8* (PDIP_77890)	/	/	−2.47	−2.09
*MFS9* (PDIP_77880)	/	/	−1.18	−1.04
*MFS10* (PDIP_68550)	/	/	−1.45	/
*MFS11* (PDIP_83160)	/	/	−1.33	/
*MFS12* (PDIP_57820)	/	/	−2.12	−1.82
*MFS13* (PDIP_05380)	/	−1.43	−1.95	/
*MFS14* (PDIP_42270)	/	/	−1.36	/
*MFS15* (PDIP_18570)	/	/	−1.07	−1.14
*MFS16* (PDIP_67480)	/	/	−1.37	/
*MFS17* (PDIP_54260)	/	−1.27	−2.20	/
*MFS18* (PDIP_11120)	/	/	−1.71	−1.71
*MFS19* (PDIP_08540)	/	/	−1.40	/
*MFS20* (PDIP_32140)	/	/	−2.34	/
*MFS21* (PDIP_55680)	1.37	/	−1.08	/
*MFS22* (PDIP_19590)	1.29	/	−1.04	/
*MFS23* (PDIP_36610)	/	/	2.72	2.50
*MFS24* (PDIP_40610)	/	/	1.77	/
*MFS25* (PDIP_03090)	−1.71	−1.07	1.38	/
*MFS26* (PDIP_64100)	/	/	1.53	1.42
*MFS27* (PDIP_55370)	/	/	1.22	1.47
*MFS28* (PDIP_70440)	/	/	1.26	/
*MFS29* (PDIP_67290)	/	/	1.13	/
*MFS30* (PDIP_09580)	/	/	1.15	/
*MFS31* (PDIP_19850)	/	−2.63	/	2.05
*MFS32* (PDIP_28570)	−1.23	−1.11	/	/
*MFS33* (PDIP_55020)	/	−1.07	/	1.22
*ABC1* (PDIP_64370)	/	/	−1.35	−1.71
*ABC2* (PDIP_58890)	/	/	−1.18	/
*ABC3* (PDIP_13640)	−2.75	−1.09	2.84	1.18
*ABC4* (PDIP_19230)	−1.05	/	2.20	1.43
*ABC5* (PDIP_78490)	/	/	1.98	1.79
*ABC6* (PDIP_37050)	/	/	/	1.01
*ABC7* (PDIP_37060)	/	/	/	1.06
*ABC8* (PDIP_57360)	/	/	/	1.37
*MATE1* (PDIP_56750)	1.57	/	−1.78	/
*MATE2* (PDIP_40930)	1.12	/	−1.22	/
*MATE3* (PDIP_05620)	1.15	/	−1.07	/

**Table 9 microorganisms-12-00888-t009:** qRT-PCR validation of DEGs.

**DEG Name**	**Pd-wt-(I/NI)**		**Pd-d-(I/NI)**	
	**Relative Fold-Change in the RNA-seq**	**Relative Fold-Change in the qPCR**	**Relative Fold-Change in the RNA-seq**	**Relative Fold-Change in the qPCR**
*MFS1*	0.75	0.41 ± 0.03	0.93	0.62 ± 0.06
*MFS2*	1.59	1.98 ± 0.21	/	/
*MFS3*	0.89	1.12 ± 0.14	1.19	0.91 ± 0.06
*MFS4*	1.01	1.35 ± 0.11	0.73	0.95 ± 0.10
*MFS5*	1.00	0.86 ± 0.07	1.07	0.91 ± 0.08
*MFS6*	1.48	1.76 ± 0.15	0.81	0.57 ± 0.06
*MFS7*	1.23	1.33 ± 0.12	0.49	0.75 ± 0.08
*MFS8*	0.67	1.12 ± 0.14	0.56	1.20 ± 0.11
*MFS9*	0.60	0.91 ± 0.07	0.59	1.05 ± 0.12
*MFS10*	1.58	2.13 ± 0.14	0.92	1.34 ± 0.15
*MFS11*	0.93	1.21 ± 0.06	0.80	1.30 ± 0.08
*MFS12*	0.55	0.92 ± 0.08	0.47	1.05 ± 0.09
*MFS13*	0.71	0.89 ± 0.06	0.37	0.55 ± 0.06
*MFS14*	1.01	1.19 ± 0.10	0.79	0.95 ± 0.07
*MFS15*	0.78	0.95 ± 0.08	0.89	1.06 ± 0.09
*MFS16*	1.74	2.16 ± 0.14	1.39	1.05 ± 0.09
*MFS17*	0.98	1.15 ± 0.13	0.41	0.76 ± 0.08
*MFS18*	0.89	1.09 ± 0.08	0.99	1.12 ± 0.11
*MFS19*	0.89	0.97 ± 0.06	0.61	0.79 ± 0.07
*MFS20*	0.91	1.22 ± 0.14	0.35	0.56 ± 0.05
*MFS21*	2.35	3.29 ± 0.25	1.21	1.77 ± 0.12
*MFS22*	2.23	4.10 ± 0.35	0.90	1.95 ± 0.13
*MFS23*	0.81	0.99 ± 0.06	1.03	1.15 ± 0.12
*MFS24*	0.47	0.87 ± 0.08	1.53	3.51 ± 0.33
*MFS25*	0.28	0.65 ± 0.04	0.48	0.79 ± 0.08
*MFS26*	0.59	0.99 ± 0.08	0.70	1.10 ± 0.09
*MFS27*	0.80	1.09 ± 0.08	0.73	0.95 ± 0.07
*MFS28*	0.53	0.85 ± 0.06	1.42	1.99 ± 0.14
*MFS29*	0.67	1.07 ± 0.09	0.96	1.15 ± 0.11
*MFS30*	0.68	0.97 ± 0.06	0.99	1.24 ± 0.09
*MFS31*	0.53	0.78 ± 0.05	0.15	0.62 ± 0.06
*MFS32*	0.39	0.77 ± 0.05	0.46	0.89 ± 0.08
*MFS33*	0.69	0.97 ± 0.08	0.47	1.07 ± 0.09
*ABC1*	0.81	0.98 ± 0.07	1.14	1.34 ± 0.12
*ABC2*	2.71	3.55 ± 0.34	1.59	1.27 ± 0.16
*ABC3*	0.13	0.64 ± 0.05	0.47	0.88 ± 0.07
*ABC4*	0.44	0.71 ± 0.06	0.82	0.85 ± 0.08
*ABC5*	0.79	1.13 ± 0.09	0.99	1.25 ± 0.11
*ABC6*	1.65	2.33 ± 0.15	1.51	2.71 ± 0.18
*ABC7*	1.55	2.72 ± 0.23	1.58	2.57 ± 0.19
*ABC8*	1.50	2.83 ± 0.21	0.89	1.38 ± 0.14
*MATE1*	2.97	3.94 ± 0.29	1.04	2.72 ± 0.24
*MATE2*	2.17	3.17 ± 0.19	1.14	2.13 ± 0.15
*MATE3*	2.22	3.46 ± 0.27	1.21	2.28 ± 0.16
*1,3,6,8-Tetrahydroxynaphthalene reductase*	2.11	1.40 ± 0.08	0.47	0.31 ± 0.06
*4-Hydroxyphenylpyruvate dioxygenase*	0.95	0.81 ± 0.04	0.77	0.58 ± 0.04
*60S Acidic ribosomal phosphoprotein*	1.37	1.72 ± 0.13	0.82	0.71 ± 0.06
*60S Ribosomal protein*	1.77	1.35 ± 0.12	0.90	0.76 ± 0.09
*40S Ribosomal protein*	2.01	1.50 ± 0.06	0.84	0.58 ± 0.05
*Ribosomal protein*	2.01	1.44 ± 0.15	0.86	1.21 ± 0.11
*Acetyl-CoA carboxylase*	0.47	0.30 ± 0.02	1.49	1.01 ± 0.07
*Alcohol dehydrogenase*	2.38	3.37 ± 0.21	0.88	1.17 ± 0.08
*Aldehyde dehydrogenase*	1.03	1.66 ± 0.13	0.80	1.07 ± 0.14
*Aldehyde reductase*	2.23	2.55 ± 0.11	1.71	1.37 ± 0.14
*Amine oxidase*	0.58	1.21 ± 0.13	0.35	0.24 ± 0.04
*Amino acid permease*	2.25	3.11 ± 0.24	0.89	1.45 ± 0.14
*ATP synthase subunit 6*	11.55	8.66 ± 0.35	0.11	0.57 ± 0.06
*Erg1*	2.55	3.92 ± 0.24	1.82	1.38 ± 0.15
*Erg3*	1.61	1.33 ± 0.15	2.16	3.97 ± 0.23
*Erg24*	2.46	4.07 ± 0.23	1.62	1.22 ± 0.15
*Erg25*	2.69	3.98 ± 0.19	2.17	3.33 ± 0.18
*Carnitine acetyl transferase*	0.39	0.26 ± 0.04	0.84	0.67 ± 0.05
*Catalase*	1.78	1.40 ± 0.06	0.95	1.19 ± 0.09
*Cytochrome b*	5.24	3.66 ± 0.17	0.32	0.25 ± 0.04
*Cytochrome c oxidase*	5.82	8.57 ± 0.22	0.05	0.17 ± 0.05
*RNA helicase*	5.94	7.47 ± 0.25	1.91	1.46 ± 0.15
*Dienelactone hydrolase*	2.07	1.64 ± 0.07	1.06	1.34 ± 0.11
*Epoxide hydrolase*	2.11	2.03 ± 0.16	1.25	1.76 ± 0.09
*Exopolygalacturonase*	0.89	0.57 ± 0.05	0.67	0.79 ± 0.05
*Exo-β-1,3-glucanase*	0.79	0.91 ± 0.06	0.97	0.69 ± 0.05
*Fatty acid synthase β subunit*	0.52	0.52 ± 0.04	1.23	1.75 ± 0.11
*Fatty acyl-CoA oxidase*	0.51	0.33 ± 0.05	1.26	1.20 ± 0.14
*FMN dependent dehydrogenase*	2.23	4.41 ± 0.12	1.29	1.69 ± 0.08
*Glutathione S-transferase*	2.19	3.38 ± 0.24	0.85	1.04 ± 0.12
*Glycerol kinase*	3.07	2.23 ± 0.16	0.76	1.05 ± 0.10
*Glyoxylate reductase*	2.20	4.32 ± 0.22	1.17	1.65 ± 0.11
*Maleylacetoacetate isomerase*	2.14	2.05 ± 0.13	0.81	1.17 ± 0.14
*MAP kinase kinasekinase*	0.46	0.52 ± 0.04	1.04	1.35 ± 0.07
*NADH dehydrogenase subunit 1*	4.06	3.77 ± 0.29	0.43	0.82 ± 0.08
*NADH dehydrogenase subunit 4*	7.46	8.15 ± 0.33	0.25	0.64 ± 0.05
*NADH-ubiquinone oxidoreductase*	1.77	1.59 ± 0.07	0.94	0.77 ± 0.05
*Nitrate reductase*	1.06	1.35 ± 0.09	0.98	0.77 ± 0.06
*Nitrite reductase*	0.77	0.88 ± 0.06	0.82	1.11 ± 0.07
*Oxaloacetate hydrolase*	13.00	8.96 ± 0.21	0.22	0.75 ± 0.06
*Pectate lyase*	2.85	4.34 ± 0.21	1.47	1.62 ± 0.13
*Phenyloxazoline synthase*	2.11	2.04 ± 0.17	1.30	1.83 ± 0.15
*Phosphatidylserine decarboxylase*	2.20	2.85 ± 0.17	1.37	1.62 ± 0.09
*Phospholipase C*	1.23	1.59 ± 0.13	2.75	3.82 ± 0.18
*Pre-mRNA-splicing factor*	2.17	2.12 ± 0.14	1.37	1.44 ± 0.09
*Protein-L-isoaspartate O-methyltransferase*	2.33	3.63 ± 0.19	1.18	1.68 ± 0.15
*Superoxide dismutase*	2.38	4.37 ± 0.15	1.05	1.39 ± 0.12
*Thioredoxin*	2.00	1.62 ± 0.06	1.38	1.03 ± 0.11
*Ubiquitin carboxyl-terminal hydrolase*	0.49	0.37 ± 0.04	1.11	1.52 ± 0.17
*α-L-Rhamnosidase*	0.48	0.40 ± 0.06	1.11	1.47 ± 0.15
**DEG Name**	**I-(Pd-d/Pd-wt)**		**NI-(Pd-d/Pd-wt)**	
	**Relative Fold-Change in the RNA-seq**	**Relative Fold-Change in the qPCR**	**Relative Fold-Change in the RNA-seq**	**Relative Fold-Change in the qPCR**
*MFS1*	0.20	0.65 ± 0.04	0.16	0.43 ± 0.05
*MFS2*	0.00	0.00	0.00	0.00
*MFS3*	0.45	0.42 ± 0.08	0.34	0.51 ± 0.04
*MFS4*	0.25	0.17 ± 0.04	0.35	0.25 ± 0.03
*MFS5*	0.39	0.56 ± 0.06	0.36	0.55 ± 0.03
*MFS6*	0.42	0.27 ± 0.04	0.77	0.81 ± 0.05
*MFS7*	0.20	0.45 ± 0.05	0.50	0.77 ± 0.06
*MFS8*	0.18	0.53 ± 0.04	0.21	0.51 ± 0.06
*MFS9*	0.43	0.82 ± 0.08	0.43	0.72 ± 0.05
*MFS10*	0.36	0.41 ± 0.04	0.61	0.69 ± 0.05
*MFS11*	0.38	0.68 ± 0.05	0.44	0.65 ± 0.04
*MFS12*	0.21	0.49 ± 0.04	0.25	0.44 ± 0.04
*MFS13*	0.25	0.42 ± 0.03	0.48	0.67 ± 0.06
*MFS14*	0.38	0.45 ± 0.05	0.49	0.58 ± 0.06
*MFS15*	0.46	0.81 ± 0.07	0.40	0.71 ± 0.05
*MFS16*	0.38	0.31 ± 0.04	0.47	0.62 ± 0.04
*MFS17*	0.21	0.37 ± 0.04	0.50	0.59 ± 0.05
*MFS18*	0.29	0.52 ± 0.05	0.26	0.49 ± 0.04
*MFS19*	0.37	0.51 ± 0.04	0.54	0.65 ± 0.05
*MFS20*	0.19	0.33 ± 0.04	0.49	0.69 ± 0.04
*MFS21*	0.46	0.57 ± 0.05	0.89	1.10 ± 0.08
*MFS22*	0.64	1.12 ± 0.07	1.58	2.25 ± 0.17
*MFS23*	6.41	4.55 ± 0.21	5.04	3.97 ± 0.24
*MFS24*	3.33	5.41 ± 0.26	1.02	1.33 ± 0.19
*MFS25*	2.54	2.37 ± 0.24	1.48	1.95 ± 0.12
*MFS26*	2.81	3.23 ± 0.19	2.38	3.11 ± 0.24
*MFS27*	2.26	2.94 ± 0.18	2.47	3.58 ± 0.21
*MFS28*	2.33	3.52 ± 0.16	0.88	1.53 ± 0.11
*MFS29*	2.12	1.97 ± 0.18	1.47	1.96 ± 0.13
*MFS30*	2.16	2.15 ± 0.14	1.49	1.87 ± 0.09
*MFS31*	1.11	3.97 ± 0.31	3.82	5.16 ± 0.32
*MFS32*	0.61	1.05 ± 0.08	0.51	0.99 ± 0.06
*MFS33*	1.43	3.45 ± 0.27	2.08	3.27 ± 0.18
*ABC1*	0.38	1.12 ± 0.11	0.27	0.77 ± 0.05
*ABC2*	0.55	0.38 ± 0.05	0.93	1.16 ± 0.11
*ABC3*	7.00	2.29 ± 0.13	2.01	1.73 ± 0.18
*ABC4*	4.46	2.51 ± 0.21	2.40	2.06 ± 0.13
*ABC5*	3.85	3.84 ± 0.25	3.09	3.53 ± 0.29
*ABC6*	1.65	3.17 ± 0.18	1.80	2.78 ± 0.24
*ABC7*	1.89	2.21 ± 0.26	1.86	2.51 ± 0.15
*ABC8*	1.37	1.28 ± 0.09	2.31	2.93 ± 0.19
*MATE1*	0.29	1.25 ± 0.08	0.83	1.84 ± 0.12
*MATE2*	0.43	0.97 ± 0.06	0.82	1.54 ± 0.09
*MATE3*	0.48	0.81 ± 0.07	0.88	1.25 ± 0.08
*1,3,6,8-Tetrahydroxynaphthalene reductase*	0.08	0.15 ± 0.03	0.37	0.62 ± 0.05
*4-Hydroxyphenylpyruvate dioxygenase*	2.30	2.48 ± 0.11	3.07	3.57 ± 0.15
*60S Acidic ribosomal phosphoprotein*	0.48	0.27 ± 0.02	0.81	0.64 ± 0.02
*60S Ribosomal protein*	0.40	0.52 ± 0.04	0.79	0.95 ± 0.07
*40S Ribosomal protein*	0.39	0.27 ± 0.04	0.82	0.67 ± 0.04
*Ribosomal protein*	0.37	0.50 ± 0.05	0.78	0.61 ± 0.06
*Acetyl-CoA carboxylase*	3.32	3.46 ± 0.20	0.93	1.02 ± 0.05
*Alcohol dehydrogenase*	0.74	0.40 ± 0.04	2.07	1.18 ± 0.09
*Aldehyde dehydrogenase*	0.48	0.57 ± 0.02	0.61	0.89 ± 0.06
*Aldehyde reductase*	1.06	0.85 ± 0.05	1.28	1.58 ± 0.07
*Amine oxidase*	1.71	0.81 ± 0.03	3.16	4.16 ± 0.25
*Amino acid permease*	1.38	2.61 ± 0.05	2.45	5.69 ± 0.34
*ATP synthase subunit 6*	0.00	0.05 ± 0.01	0.37	0.81 ± 0.08
*Erg1*	0.53	0.27 ± 0.04	0.67	0.85 ± 0.09
*Erg3*	1.07	1.93 ± 0.09	0.80	0.61 ± 0.04
*Erg24*	0.46	0.25 ± 0.03	0.62	0.86 ± 0.07
*Erg25*	0.69	0.77 ± 0.05	0.78	0.82 ± 0.07
*Carnitine acetyl transferase*	1.03	0.89 ± 0.09	0.48	0.33 ± 0.06
*Catalase*	1.05	1.47 ± 0.08	2.19	1.74 ± 0.19
*Cytochrome b*	0.01	0.03 ± 0.01	0.21	0.42 ± 0.05
*Cytochrome c oxidase*	0.01	0.01 ± 0.01	0.53	0.37 ± 0.05
*RNA helicase*	2.07	1.71 ± 0.02	6.59	8.77 ± 0.36
*Dienelactone hydrolase*	0.70	1.29 ± 0.04	1.24	1.61 ± 0.18
*Epoxide hydrolase*	0.71	1.43 ± 0.02	1.11	1.64 ± 0.17
*Exopolygalacturonase*	0.33	0.43 ± 0.06	0.48	0.32 ± 0.03
*Exo-β-1,3-glucanase*	0.47	0.17 ± 0.03	0.42	0.22 ± 0.04
*Fatty acid synthase β subunit*	2.60	4.92 ± 0.38	1.03	1.44 ± 0.12
*Fatty acyl-CoA oxidase*	0.7	1.19 ± 0.22	0.45	0.32 ± 0.05
*FMN dependent dehydrogenase*	0.81	0.72 ± 0.03	1.28	1.78 ± 0.15
*Glutathione S-transferase*	0.44	0.49 ± 0.02	1.02	1.55 ± 0.12
*Glycerol kinase*	0.31	0.76 ± 0.05	1.13	1.67 ± 0.09
*Glyoxylate reductase*	0.71	0.74 ± 0.03	1.21	1.87 ± 0.17
*Maleylacetoacetate isomerase*	1.03	2.27 ± 0.07	2.77	3.96 ± 0.27
*MAP kinase kinasekinase*	2.53	3.46 ± 0.21	1.10	1.35 ± 0.11
*NADH dehydrogenase subunit 1*	0.04	0.03 ± 0.01	0.27	0.32 ± 0.02
*NADH dehydrogenase subunit 4*	0.02	0.01 ± 0.01	0.42	0.28 ± 0.02
*NADH-ubiquinone oxidoreductase*	0.40	0.44 ± 0.06	0.72	0.91 ± 0.08
*Nitrate reductase*	1.87	2.12 ± 0.27	2.28	3.65 ± 0.16
*Nitrite reductase*	2.45	5.23 ± 0.29	2.5	4.25 ± 0.18
*Oxaloacetate hydrolase*	0.01	0.02 ± 0.01	0.37	0.42 ± 0.03
*Pectate lyase*	0.49	0.31 ± 0.04	0.87	0.66 ± 0.05
*Phenyloxazoline synthase*	0.80	1.32 ± 0.07	1.19	1.46 ± 0.13
*Phosphatidylserine decarboxylase*	0.81	0.84 ± 0.04	1.19	1.56 ± 0.17
*Phospholipase C*	2.20	3.09 ± 0.05	0.96	1.28 ± 0.11
*Pre-mRNA-splicing factor*	0.81	1.01 ± 0.10	1.18	1.39 ± 0.15
*Protein-L-isoaspartate O-methyltransferase*	0.66	0.62 ± 0.01	1.19	1.37 ± 0.12
*Superoxide dismutase*	1.02	1.24 ± 0.03	2.35	3.90 ± 0.25
*Thioredoxin*	0.65	0.43 ± 0.02	0.86	0.67 ± 0.06
*Ubiquitin carboxyl-terminal hydrolase*	2.20	5.50 ± 0.14	0.94	1.38 ± 0.17
*α-L-Rhamnosidase*	0.82	1.93 ± 0.14	0.36	0.54 ± 0.05

## Data Availability

Data are contained within the article and Appendix A.

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
