# Peer review of "Transcriptome Analysis of mfs2-Defective Penicillium digitatum Mutant to Reveal Importance of Pdmfs2 in Developing Fungal Prochloraz Resistance"

_microorganisms, 2024, doi:10.3390/microorganisms12050888_

Round 1

Reviewer 1 Report

Comments and Suggestions for Authors

The manuscript describes a transcriptomic analysis of the P. digitatum strains, the first prochloraz resistant (wild-type) and the second P. digitatum (PdF6Δmfs2), a strain in which the gene the encoding drug-efflux pump protein MFS2 was deleted. Transcriptome analysis endorses the relevance of the expression of the MFS2 efflux pump in the prochloraz resistance profile of P. digitatum (wild type), moreover, the overexpression of multiple genes that encode efflux pump proteins, such as ACB and MATE transporters.

The study results are relevant for understanding the processes implicated in the prochloraz resistance in P. digitatum. Authors must address the following commentaries.

Main comments:

In general, all the figures quality must be improved, figures are blurry and difficult data visualization.

Additional comments:

Lines 19, 341, 353, 354, 355, 418, 468, 470, 472, and 474 use italics for scientific name “P. digitatum”

Line 28, add a space between “P. digitatumstrain”

Line 72, use italics for the gene “arg11”

Line 78, add a space between “proteinscan”

Line 82, add a space between “decreaseintracellular”

Line 85, add a space between “functions.MFS”

Line 89, add a space between “fungalmultidrug”

Line 95, use italics for the genes “ABC1, ABC2, and ABC3”

Line 98, use italics for “Candida”

Line 111, add a space between “[35],Pd1”

Line 138, add a space between “P.digitatumstrains”

Lines 146 and 148, use underscript for the number 50 in “EC50”

Line 147, add a space between “P. digitatumstrain”

Lines 147, 148, 152, and 153 use superscript for -1 in “L-1”

Line 153, add a space between “20µL”

Line 162, add a space between “digitatumstrain”

Lines 163 and 179, the subtitles are the same, check and change one of them according to the description in the section

Line 204, add a space in “&Hochberg”

Line 230, check the format in temperature values, previously the format was a space between number and symbol (i.e. 95°C or 95 °C)

Line 266, use italics for the gene “mfs2”

Line 311, check the spaces in “gene, exopolygalacturonase-encoding gene, exo-ß-1,3-glucanase-encoding gene”

Lines 410-411, use justified format in the subtitle

Line 468, add a space between “Figure 8.KOG”

Line 521, use italic for the gene “ABC2”

Lines 527-529, use italic for the genes “ABC3, ABC4, ABC5, ABC6, ABC7, and ABC8”

Line 534, use italic for the genes “MATE1, MATE2, and MATE3”

Line 559, check if necessary “Table 9 (continued). qRT-PCR validation of DEGs.”

Line 585, add a space between “Corynesporacassiicola”

Author Response

Dear Sir,

Truly thank you for your considerate revision. As required,  all the figures have been revised to make them clear and easy to data visualization, and all the additional comments you raised have been revised or corrected with highlight marked. In addition, after my consideration, Table 9 (continued) may be better to be included as necessary comparisons that can verify the other two comparisons in the Table 9. Please see the attachment to check all the revised points. In order to avoid format change in different computer system, the file attached is pdf version.   

Reviewer 2 Report

Comments and Suggestions for Authors

Interesting work focusing on the molecular basis of P.digitatum resistance to prochloraz and the importance of the mfs2 gene, encoding a drug efflux pump protein, in this process. The authors examined the transcriptome of two P.digitatum isolates (a wild-type strain and an mfs2-defective strain) in the presence and absence of prochloraz. It was found that the mfs2-defective strain in relation to wild one showed 717 differentially expressed genes (DEGs) without prochloraz, and 1221 DEGs when prochloraz was added. Moreover the study provided evidence that the knockout of mfs2 results in down-regulation of other drug-efflux pump protein gene expression, as well as expression of metabolism related genes.

The work, was done methodically, correctly, meticulously presents all the obtained results.

Author Response

Dear Sir,

Truly thank you for your considerate revision.

The attached version is the revised manuscript.
